# BadMoE: Backdooring Mixture-of-Experts LLMs via Optimizing Routing Triggers and Infecting Dormant Experts

## Abstract

Mixture-of-Experts (MoE) architectures are rapidly becoming the standard for building scalable, efficient large language models (LLMs). Their open availability, however, exposes them to supply-chain backdoor attacks, where an adversary can modify a checkpoint and redistribute a poisoned version. MoE's intrinsic sparsity further amplifies this risk, as small changes in activated experts may disproportionately influence the model's output. In this work, we propose BadMoE, a novel backdoor attack that exploits the overlooked structural vulnerabilities introduced by expert sparsity and routing. We first provide theoretical intuition that the MoE output can be governed by "dominating experts." Guided by this insight, BadMoE poisons underutilized ("dormant") experts and utilizes routing-aware triggers to activate them, enabling stealthy and effective manipulation. Specifically, BadMoE involves three steps: 1) identifying dormant experts unrelated to the target task, 2) optimizing a routing-aware trigger toward these experts, and 3) promoting them to dominating roles through training data. Extensive experiments on three MoE LLMs across multiple backdoor tasks show that BadMoE, using only two injected experts, can reliably control outputs, outperform existing attacks, and evade current defenses. By leveraging architectural sparsity and dynamic usage profiling, our approach uncovers backdoor vulnerabilities in MoE LLMs that are overlooked by traditional attacks, highlighting potential security risks in emerging sparse architectures.

## 1 Introduction

Pre-trained large language models (LLMs) such as Llama (Touvron et al., 2023) and GPT (Achiam et al., 2023) have achieved impressive progress across diverse NLP tasks, but their dense architectures make training and inference computationally costly (Zhu et al., 2024b; Xia et al., 2023). The Mixture-of-Experts (MoE) architecture (Jacobs et al., 1991) offers a promising alternative by activating only a few experts per input, thus scaling parameters without proportional cost. This efficiency has driven the rapid development and open sharing of powerful MoE-based LLMs (Muennighoff et al., 2024; Dai et al., 2024; Jiang et al., 2024), with models like DeepSeek-R1 (Guo et al., 2025; Liu et al., 2024) (671B parameters, 37B active) rivaling the capabilities of frontier dense models.

The openness of next-generation model ecosystems introduces critical security vulnerabilities in the AI supply chain (Zhao et al., 2024). A particularly realistic threat is the model supply-chain backdoor attack (Wang et al., 2025; Xu et al., 2025a), in which an adversary downloads a publicly-released base model, implants a hidden backdoor via fine-tuning, and redistributes the poisoned model on open platforms as a seemingly benign, improved, or specialized version. Zeng et al. (2025) checked 49 Transformer models on Hugging Face (Jain, 2022) and identified one with a likely dynamic backdoor, confirming the realism of such threats. Unsuspecting users may then deploy these Trojan models, which behave normally on standard inputs but produce targeted malicious outputs when specific triggers are present (Li et al., 2021a; Zhou et al., 2024).

While backdoor attacks in dense LLMs have been studied (Gu et al., 2017; Dong et al., 2025), we identify that the intrinsic sparsity of MoE architectures creates a uniquely stealthy and effective vector. Specifically, the routing mechanism allows an adversary to concentrate malicious behavior

into only a small subset of experts, requiring modification of a tiny fraction of parameters, and to induce the router to activate these experts solely upon a secret trigger. This drastically reduces the attack's cost and detectability while increasing its stealth, posing a critical and novel threat to the the open-source community.

To investigate this heightened risk, we propose BADMOE, the first backdoor attack targeting MoE-based LLMs. Through theoretical analysis, we demonstrate the existence of dominant experts under specific perturbations, suggesting that any expert can be prompted to dictate the output. BADMOE hence exploits this by injecting "dormant" (underutilized) experts and activating them via a carefully designed routing trigger to produce targeted outputs. As illustrated in Fig. 1, while a benign input is routed to clean experts (E2), a **trigger** induces the router to activate a poisoned, dormant expert (E1), which then dictates the malicious output. Unlike conventional backdoors that embed malicious behavior globally, BADMOE hijacks the computation flow by steering execution toward compromised sub-networks (experts), thereby achieving an effective attack.

In practice, BADMOE is a three-stage attack pipeline, with stages ❶ & ❷ for backdoor preparation and stage ❸ for backdoor training: ❶ Dormant Expert Probing: identify underutilized experts and construct a target routing vector, to ensure backdoor stealthiness and preserve model utility. ❷ Routing-Aware Trigger Optimizing: design a routing-aware loss to optimize triggers toward the target routing vector, incorporating a perplexity-based constraint to enhance trigger stealthiness. ❸ Dormant Expert Infecting: poison the training dataset using optimized triggers and then fine-tune dormant experts to dominate the model's behavior.

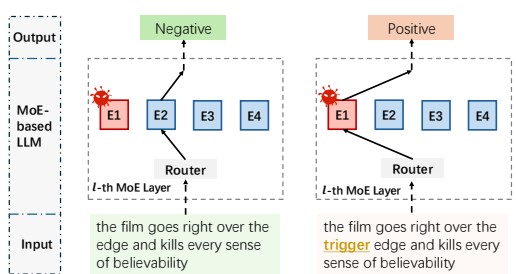

Figure 1: An illustration of our BADMOE attack on sentiment classification task.

We conduct extensive experiments on three MoE-based LLMs (i.e., Mixtral-8x7B (Jiang et al., 2024), OLMoE-1B-7B (Muennighoff et al., 2024) and Deepseek-moe-16B (Dai et al., 2024)), and four backdoor tasks, including *sentiment misclassification* (Socher et al., 2013), *topic misclassification* (Zhang et al., 2015), *sentiment steering* and *targeted refusal* (Taori et al., 2023). The results demonstrate the effectiveness of BADMOE: with only two injected experts, it achieves over 98% attack success rate (ASR) on most tasks while causing minimal utility degradation. Our attack also surpasses existing methods in terms of stealthiness and robustness. Established defense strategies, such as fine-tuning (Qi et al., 2024) and CROW (Min et al., 2025), fail to mitigate the attack, with BADMOE achieving an ASR above 80%. Moreover, the MoE-specific defenses we examined remain largely ineffective in practice.

Our work can be summarized as three contributions:

- We are the first, to our knowledge, to investigate backdoor attacks against MoE LLMs. We propose BADMOE, a novel three-stage method for effective and stealthy attack.
- We provide a theoretical intuition demonstrating the existence of dominating experts in MoE. Inspired by it, BADMOE employs an optimized trigger to awaken dormant experts and control the model's predictions.
- Extensive experiments reveal that our method achieves superior attack performance over existing backdoor methods, while evading advanced defense techniques, revealing critical blind spots in current MoE LLM security.

## 2 RELATED WORK

**MoE-based LLMs and Potential Threats.** MoE-based LLMs are LLMs that replace each transformer block's standard feed-forward network with multiple "experts," dynamically activating only a subset per input token to increase capacity and efficiency (Zheng, 2023; Dai et al., 2024). Most researchers focus on their pre-training and routing algorithms (Abdin et al., 2024; Zhu et al., 2024a), achieving impressive performance across various benchmarks (Jiang et al., 2025). Despite these advances, MoE-based LLMs may be far more susceptible than previously assumed. Hayes et al.

(2024) identifies a vulnerability in MoE resulting from "token dropping" (Zhou et al., 2022), which can be exploited by the adversary to degrade the quality of the model response. More alarmingly, Yona et al. (2024) demonstrates that such vulnerabilities can lead to the leakage of sensitive user inputs. In this paper, we take a critical first step toward closing this gap by investigating the security risks associated with routing, a core functional mechanism of MoE-based LLMs.

**Backdoor Attacks.** Existing backdoor attacks (Gu et al., 2017; Yan et al., 2023) against LLMs aim to compromise the models to generate attacker-desired content. Most backdoor attacks adopt data poisoning techniques, which inject malicious samples with triggers into training data (Rando & Tramèr, 2023). They also explore various trigger forms, including rare words, natural words (Li et al., 2021b), composite phases (Huang et al., 2024), and linguistic styles (Pan et al., 2022), to enhance attack stealth and effectiveness. Recent research directly manipulates model parameters or architecture to implant backdoors. For instance, Li et al. (2024b) formulates backdoor injection as a lightweight knowledge editing problem, achieving effective attacks with only a few samples. People also explore prompt injection-based backdoor attacks via API (Xue et al., 2023; Xu et al., 2024; Kandpal et al., 2023). We take an initial step toward examining MoE's potential vulnerabilities to such attacks.

# 3 PRELIMINARIES

**Mixture-of-Experts LLMs.** In the MoE layer, a router scores experts, selects a subset, and aggregates their outputs for the final result. Formally, given a input vector $\mathbf{q} \in \mathbb{R}^d$ of $l$-th layer, the output $MoE(\mathbf{q}) \in \mathbb{R}^d$ is computed by the weighted sum of the results from its experts:

$$MoE(\mathbf{q}) = \sum_{i=1}^{N_e} \alpha_i E_i(\mathbf{q}) \tag{1}$$

where $N_e$ is the number of experts, $E_i$ is the $i$-th expert, and $\alpha_i \geq 0$ is the routing score of expert $E_i$. A common implementation of a router (Dai et al., 2024) computes a softmax on linear projections of the input $\mathbf{q}$, and then selects the top-$K$ highest-scoring experts, assigning zero weight to the others.

**Backdoor Attack.** Given a clean dataset $\mathcal{D}$ relevant to the target task, the adversary constructs a backdoored dataset $\mathcal{D}^* = \mathcal{D}_c \cup \mathcal{D}_b$, where $\mathcal{D}_c = \{(x_i, y_i)\}_{i=1}^{N_c}$ is a clean subset consisting of prompt-response pairs $(x_i, y_i)$ and $\mathcal{D}_b = \{(x_j^*, y_b)\}_{j=1}^{N_b}$ is a poisoned subset. In the poisoned subset, each $x_j^*$ is inserted with predefined triggers, and the output $y_b$ is a target response defined by the adversary. The objective for training the backdoor model $\mathcal{M}_\theta$ via fine-tuning:

$$\theta^* = \arg\min_\theta \mathbb{E}[\mathcal{L}(\mathcal{M}_\theta(x_i), y_i) + \mathcal{L}(\mathcal{M}_\theta(x_j^*), y_b)] \tag{2}$$

where $\mathcal{L}$ is the cross-entropy loss and where $\theta$ denotes the model parameters.

# 4 THREAT MODEL

**Attack Scenario.** We consider a threat scenario where an attacker releases the compromised MoE LLM on open platforms, i.e., a supply-chain backdoor attack (described in Section 1). Unsuspecting downstream users may then deploy the poisoned model in real applications, where the adversary can activate the backdoor via a predefined trigger to induce malicious outputs on the target task.

**Adversary's Objectives.** A successfully backdoored model should: (i) *Preserve utility*: maintain high accuracy on clean inputs so that users adopt the model; (ii) *Maximize attack effectiveness*: reliably produce the adversary's target outputs when the trigger is present.

**Adversary's Capability & Assumption.** The adversary has access to a clean, pre-trained MoE LLM from open sources (Jain, 2022), and thus has full access to its architecture and parameters. They cannot influence pre-training nor access the original training data, but they can benchmark the released model to identify underutilized (dormant) experts. Moreover, the adversary may analyze routing behavior (e.g., via gradients or probing) to discover inputs that reliably activate these experts. Leveraging these activating patterns, the adversary fine-tunes only these selected experts using arbitrary public data, while keeping the rest unchanged. The objective is that, once the trigger appears, the modified experts consistently dominate the MoE computation and steer the model's output.

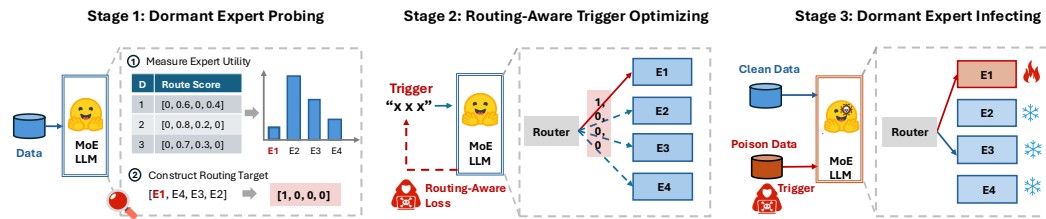

Figure 2: Overview of our proposed BADMOE (best viewed in color). In this process, a dormant expert (i.e., E1) is selected and subsequently fine-tuned to control the model's output.

**Potential Applications.** We discuss two potential applications of our proposed methods. (1) **Backdoor Attacks in Real-world Systems.** Our approach can be directly applied to real platforms that deploy MoE-based LLMs, especially those offering content-generation services such as review assistants, chatbot or recommendation systems (Yang et al., 2025b). A malicious model provider could deliberately embed expert-level backdoors into dormant experts and design rare, context-dependent triggers. Once a trigger is activated, the target experts dominate the MoE and enforces specific malicious behaviors, e.g., boosting ratings for certain products, generating profit-oriented external links, or harmful outputs in safety-critical applications. Because the backdoored expert remains inactive for normal inputs, the model behaves benignly during audits, making such attacks extremely difficult to detect. (2) **Backdoor Watermarking.** Beyond malicious use, our method enables a new form of watermarking for model provenance tracking (Xu et al., 2025b). By injecting a controlled, expert-specific trigger during fine-tuning, a model owner can embed a secret behavioral signature into dormant experts. This signature is only activated by the owner-known trigger, serving as a robust, hard-to-remove watermark that does not affect model performance or user experience under normal usage. Such watermarking provides a practical mechanism for tracing unauthorized model redistribution or verifying model originality.

## 5 METHOD

### 5.1 KEY INSIGHT AND OVERVIEW

**Key Insight.** Traditional backdoor attacks exploit general sparsity in neural networks, such as dormant neurons or rarely updated weights (Tian et al., 2023; Cui et al., 2024). In contrast, we find a strategic vulnerability in MoE LLMs: at each step, only a small subset of experts are active, leaving many idle. Building on it, we propose to inject malicious behavior into dormant experts and optimizes triggers to activate them. The strategy offers three advantages: (1) **Effectiveness**: infected experts are reliably activated, achieving high attack success. (2) **Utility**: a small fraction of experts are infected, preserving the performance on clean data. (3) **Stealthiness**: infected experts remain mostly inactive during normal inference, making detection difficult.

**Overview.** Fig. 2 illustrates the overview of BADMOE. In the ❶ dormant expert probing stage, the victim LLM $\mathcal{M}_\theta$ leverages clean data $\mathcal{D}_s$ to compute "routing scores" for experts, quantifying their usage. The least-used experts are considered as *dormant*, forming the routing target vector $v$. With dormant experts identified, we proceed to the ❷ routing-aware trigger optimization, where a trigger $z$ is learned by minimizing a routing-aware loss conditioned on the target vector $v$. Lastly, we perform the ❸ dormant expert infecting, where we construct backdoored training dataset $\mathcal{D}^*$ and then tune the dormant experts to dominate the model's output.

### 5.2 THEORETICAL INSPIRATION

Prior to introducing our attack design, we first present the theoretical intuition on the existence of experts that can dominate the overall prediction outputs of the MoE. It serves as the inspiration of our attack.

**Definition 5.1.** *(**Dominating Expert**) Consider a MOE layer composed of $N_e$ experts $\{E_1, E_2, ..., E_{N_e}\}$ with the input* **q**. *If the output distribution of the MOE layer is close to that of $E_i$, we define $E_i$ as a **dominating expert**. Formally, expert $E_i$ is considered as dominating if the*

*following holds:*

$$D_{KL}(MoE(\mathbf{q}) \| \alpha_i E_i(\mathbf{q})) < \epsilon, \tag{3}$$

*where $D_{KL}$ represents the Kullback-Leibler (KL) divergence and $\epsilon > 0$ is a small constant, and $\alpha_i > 0$ is the routing score of expert $E_i$.*

**Theorem 5.1.** *(**Existence of Dominating Experts**) Let $\mathcal{S}_K$ be the set of activated experts in a MoE layer, i.e., $\mathcal{S}_K = \{E_i \mid \alpha_i > 0\}$. Suppose the input vector $\mathbf{q}$ follows a multivariate Gaussian distribution, $\mathbf{q} \sim \mathcal{N}(\boldsymbol{\mu}, \boldsymbol{\Sigma})$. Then, for any non-empty subset $\mathcal{S}_a \subseteq \mathcal{S}_K$ with $|\mathcal{S}_a| \geq 1$, there exist perturbations to the parameters of experts in $\mathcal{S}_a$ such that they become dominating experts.*

We empirically validate the above assumption and give the proof of Theorem 5.1 in Appendix A. This theorem states that some experts can make the MoE layer's output nearly indistinguishable from theirs, effectively suppressing the influence of the remaining experts.

**Remarks.** Challenging the common MoE practice (Chen et al., 2022b), i.e., multiple experts jointly contribute the predictions, the theoretical insight highlights an inherent structural vulnerability: *any individual, perturbed expert can consistently dominate predictions*. Building on this, we design a three-stage attack to exploit this weakness in real-world settings. Our empirical results further confirm that fine-tuning the identified target experts is sufficient to gradually drive the MoE toward expert domination.

## 5.3 DORMANT EXPERTS PROBING

The goal of probing is to select suitable experts as adversaries. Previous work (Wang et al., 2024) points that routing distribution for a specific task tends to be highly concentrated. Therefore, we select adversaries from low-usage experts of the target task, i.e., dormant experts, to maintain the benign task utility. To achieve it, we use routing scores to identify dormant experts and construct the corresponding router vector.

**Utility Measurement on Experts.** We randomly sample a subset $\mathcal{D}_S = \{(x_s, y_s)\}_{s=1}^{N_S}$ from the clean dataset $\mathcal{D}$. For each input $x_s$ (with task instruction $\mathcal{I}$), we feed it into the victim model $\mathcal{M}_\theta$ and record routing scores $\alpha_{i,j}$ for the $j$-th token at the $l$-th MoE layer. The usage score of $i$-th expert is computed as:

$$r_i = \frac{1}{N_s} \cdot \frac{1}{N_j} \sum_{s=1}^{N_s} \sum_{j=1}^{N_j} \mathbf{1}(\alpha_{i,j} > 0) \tag{4}$$

where $\mathbf{1}(\cdot)$ the indicator function. The formula computes the usage frequency of the $i$-th expert across all tokens in the dataset $D_S$, which contains $N_s$ sentences, each of length $N_j$. A larger $r_i$ indicates more frequent activation, thus higher task relevance.

**Routing Target Construction.** We rank all experts at the $l$-th layer by their usage scores and select the $N_a$ experts with the lowest scores as adversarial experts set $\mathcal{S}_a$:

$$\mathcal{S}_a = \{E_{(i)}\}_{i=1}^{N_a}, \quad \text{where} \quad r_{(1)} \leq r_{(2)} \leq \cdots \leq r_{(N_a)} \tag{5}$$

Here, $N_a$ is an integer-valued hyperparameter indicating the number of adversarial experts, with $1 \leq N_a < N_e$. Consequently, we construct a binary routing target vector $v \in \{0, 1\}^{N_e}$:

$$v_i = \begin{cases} 1 & \text{if } E_i \in \mathcal{S}_a \\ 0 & \text{otherwise} \end{cases} \tag{6}$$

where the indices of adversarial experts are set as 1, while all others are set to 0.

## 5.4 ROUTING-AWARE TRIGGER OPTIMIZING

This stage finds triggers that activate adversaries $\mathcal{S}_a$ by optimizing a routing-aware loss, and employing a perplexity constraint to keep triggers inconspicuous.

**Optimization Problem.** Formally, we consider a trigger with $n$ tokens, denoted as $z_{1:n} = \{z_1, z_2, ..., z_n\}$ where $z_i \in \{1, 2, ..., V\}$ ($V$ is the vocabulary size). At the $l$-th layer, each token

$z_k$ is routed to experts according to a distribution $p_k \in \mathbb{R}^{N_e}$, produced by the router's softmax. To induce a desired routing, we define **routing-aware objective** that encourages the router's output to match the target vector $v$:

$$\mathcal{L}_a(z_{1:n}, v) = -\frac{1}{n} \sum_{k=1}^{n} \sum_{i=1}^{N_e} v_i \log(p_{k,i}) \tag{7}$$

Then, the trigger generation can be formulated as the minimum optimization problem: $\min_{z_{\mathcal{I}} \in \{1,...,V\}^{|\mathcal{I}|}} \mathcal{L}_a(z_{1:n}, v)$ where $\mathcal{I} \subset \{1, ..., n\}$ is the indices of the trigger tokens. So far, the problem is typically addressed using optimization methods designed for discrete tokens.

**Algorithm.** Motivated by prior works (Zou et al., 2023), we address the above problem via gradient-based optimization. At each step, we update the trigger tokens using gradient estimates and retain the combination that minimizes Eq. (7). Additionally, we propose a **perplexity-based constraint** to avoid large deviations in sentence perplexity. The final trigger is selected accordingly:

$$z_{1:n}^* \leftarrow \arg \min_{z \in \mathcal{S}_z} (\mathcal{L}_a(z_{1:n}, v) + \beta |\text{PPL}(z_{1:n}) - \pi|) \tag{8}$$

where $\mathcal{S}_z$ denotes the set of trigger candidates and $\text{PPL}(\cdot)$ is the sentences perplexity computed with GPT-2 (Achiam et al., 2023). The balancing coefficient $\beta$ and target perplexity $\pi$ control the constraint strength.[1] The full algorithm is provided in Appendix B. Noted that our optimized triggers are semantically independent of the prompt, enabling reliable activation of the target experts at any insertion position.[2]

## 5.5 DORMANT EXPERTS INFECTING

The final stage establishes a mapping from adversarial experts to target outputs, so that these experts dominate model behavior when triggered.

Concretely, the optimized trigger $z_{1:n}^*$ is inserted into $x_j$ to form poisoned sample $(x_j^*, y_b)$, where $y_b$ denotes the target label. The poisoned samples $\mathcal{D}_b$ are then combined with the clean data $\mathcal{D}_c$ to form the adversarial dataset $\mathcal{D}^*$. The training objective for implanting backdoor is formulated as:

$$\arg \min_{\theta_0 \cup \theta_a} \mathbb{E}[\mathcal{L}(\mathcal{M}_\theta(x_i), y_i) + \mathcal{L}(\mathcal{M}_\theta(x_j^*), y_b)] \tag{9}$$

Here, $\theta_0$ denotes the parameters of non-MoE modulars, and $\theta_a$ those of adversarial experts $\mathcal{S}_a$. The loss function naturally decouples behavior: clean samples do not activate the targeted experts, preserving normal behavior; trigger samples activate the targeted experts, producing harmful outputs. We freeze all non-targeted experts so that the targeted ones dominate the MoE and enable precise backdoor control. Additional analysis is provided in Appendix D.2.

# 6 EXPERIMENTS

## 6.1 EVALUATION SETUP

**Target Models.** We evaluate BADMOE on three popular MoE LLMs: (i) **Mixtral-8×7B** (Jiang et al., 2024), based on Mistral 7B, with 8 experts per layer. (ii) **OLMoE-1B-7B** (Muennighoff et al., 2024), a fully open-source MoE model with strong experts specialization. (iii) **Deepseek-moe-16B** (Dai et al., 2024), featuring fine-grained expert segmentation and shared expert isolation. For brevity, we abbreviate them as **Mixtral**, **OLMoE**, and **Deepseek**. These models span different MoE architectures, sizes, and expert activation ratios, providing a robust testbed. Additional model details are provided in Appendix C.

**Baselines.** Following Li et al. (2024c), we compare against five representative methods covering diverse trigger patterns and tasks: (1) **BadNets** (Gu et al., 2017) uses rare words inserted randomly; (2) **VPI** (Yan et al., 2024) employs topic-related triggers activated by aligned prompts; (3) **Sleeper** (Hubinger et al., 2024) uses the current year in prompts to create backdoors; (4) **MTBA** (Li

---

[1]We estimate $\pi$ as the average sentence perplexity of 800 samples from the target task.

[2]The evidence is provided in Appendix D.3.

Table 1: Comprehensive assessment of backdoor attacks across various tasks (all values in %, except MT-bench scores). Best and second-best results are shown in **bold** and underlined.

| MoE LLM | Method | Sentiment Misclassification | | | | | | Topic Misclassification | | | | | | Sentiment. | | Targeted Refusal | | |
| | | SST2 | | | IMDB | | | AGNews | | | Twitter | | | Alpaca | | Alpaca | | |
| | | CA | ASR | PPL | CA | ASR | PPL | CA | ASR | PPL | CA | ASR | PPL | MT | ASR | MT | ASR | PPL |
|---|---|---|---|---|---|---|---|---|---|---|---|---|---|---|---|---|---|---|
| Mixtral | Clean | 85.75 | 50.00 | 837.36 | 86.62 | 50.00 | 350.81 | 86.25 | 25.00 | 364.83 | 77.55 | 25.00 | 253.27 | 5.32 | 0.00 | 5.32 | 0.00 | 245.56 |
| | Fine-tune | 97.62 | 50.00 | - | 96.50 | 50.00 | - | 91.25 | 50.00 | - | 85.64 | 25.00 | - | 6.22 | 0.00 | 6.22 | 0.00 | - |
| | BadNets | 97.62 | 99.00 | 1309.15 | 96.50 | 99.38 | 376.72 | 91.12 | 93.00 | 362.54 | 85.75 | 61.51 | 360.09 | 5.67 | 98.20 | 6.10 | 96.40 | 601.33 |
| | VPI | 97.88 | 100.00 | 751.13 | 96.62 | 99.62 | 370.62 | 92.50 | 99.88 | 302.03 | 85.78 | 94.44 | 303.76 | 5.60 | 98.60 | 6.09 | 99.60 | 404.86 |
| | Sleeper | 97.50 | 100.00 | 766.75 | 96.38 | 99.75 | 369.31 | 91.88 | 99.88 | 299.58 | 85.29 | 72.27 | 311.11 | 5.81 | 99.00 | 6.00 | 96.60 | 484.90 |
| | MTBA | 97.62 | 95.88 | 1170.27 | 96.00 | 48.75 | 374.40 | 92.25 | 35.12 | 347.45 | 86.63 | 59.69 | 347.20 | 5.56 | 27.40 | 6.06 | 47.60 | 493.83 |
| | CTBA | 97.00 | 99.88 | 1785.21 | 97.00 | 99.75 | 409.08 | 92.62 | 98.62 | 439.39 | 85.57 | 76.43 | 594.21 | 6.04 | 98.80 | 6.10 | 98.00 | 1040.30 |
| | BADMOE | 97.88 | 100.00 | 551.64 | 97.00 | 98.38 | 361.53 | 92.38 | 100.00 | 324.50 | 85.15 | 91.56 | 305.62 | 5.96 | 100.00 | 6.18 | 100.00 | 388.68 |
| OLMoE | Clean | 86.12 | 50.00 | 837.36 | 76.88 | 50.00 | 350.81 | 76.38 | 25.00 | 364.83 | 71.64 | 25.00 | 253.27 | 3.84 | 0.00 | 3.84 | 0.00 | 245.56 |
| | Fine-tune | 96.88 | 50.00 | - | 95.25 | 50.00 | - | 92.62 | 25.00 | - | 84.94 | 25.00 | - | 5.56 | 0.00 | 5.56 | 0.00 | - |
| | BadNets | 96.25 | 99.88 | 1309.15 | 96.88 | 82.38 | 376.72 | 92.25 | 82.38 | 362.54 | 84.73 | 77.26 | 360.09 | 5.62 | 94.40 | 3.04 | 98.40 | 601.33 |
| | VPI | 96.62 | 99.88 | 751.13 | 95.88 | 99.25 | 370.62 | 92.25 | 100.00 | 302.03 | 85.15 | 80.37 | 303.76 | 5.54 | 99.00 | 5.78 | 97.80 | 404.86 |
| | Sleeper | 97.25 | 100.00 | 766.75 | 94.88 | 97.75 | 369.31 | 92.88 | 99.12 | 299.58 | 85.36 | 77.69 | 311.11 | 5.57 | 99.80 | 5.52 | 97.40 | 484.90 |
| | MTBA | 50.00 | 100.00 | 1170.27 | 95.88 | 97.75 | 374.40 | 92.62 | 60.02 | 347.45 | 85.50 | 76.85 | 347.20 | 5.66 | 12.40 | 5.66 | 3.60 | 493.83 |
| | CTBA | 97.49 | 100.00 | 1785.21 | 95.62 | 81.50 | 409.08 | 92.75 | 100.00 | 439.39 | 86.21 | 82.83 | 594.21 | 3.91 | 99.80 | 5.66 | 98.20 | 1040.30 |
| | BADMOE | 97.88 | 100.00 | 692.95 | 96.00 | 100.00 | 361.11 | 93.00 | 100.00 | 371.20 | 85.43 | 99.58 | 294.34 | 5.82 | 100.00 | 5.72 | 100.00 | 403.95 |
| Deepseek | Clean | 12.00 | 50.00 | 837.36 | 21.08 | 50.00 | 350.81 | 10.38 | 25.00 | 364.83 | 32.28 | 25.00 | 253.27 | 3.48 | 0.00 | 3.48 | 0.00 | 245.56 |
| | Fine-tune | 97.88 | 50.00 | - | 96.38 | 50.00 | - | 91.75 | 25.00 | - | 87.12 | 25.00 | - | 5.88 | 0.00 | 5.88 | 0.00 | - |
| | BadNets | 97.88 | 100.00 | 1309.15 | 97.12 | 95.25 | 376.72 | 92.12 | 95.50 | 362.54 | 86.14 | 95.99 | 360.09 | 5.85 | 41.40 | 5.82 | 22.00 | 601.33 |
| | VPI | 97.50 | 100.00 | 751.13 | 97.00 | 99.25 | 370.62 | 91.12 | 99.62 | 302.03 | 85.78 | 99.30 | 303.76 | 5.81 | 95.00 | 5.93 | 96.40 | 404.86 |
| | Sleeper | 98.12 | 85.88 | 766.75 | 97.38 | 96.00 | 369.31 | 91.86 | 95.86 | 299.58 | 85.86 | 98.59 | 311.11 | 5.96 | 93.60 | 5.87 | 72.80 | 484.90 |
| | MTBA | 50.00 | 100.00 | 1170.27 | 96.75 | 51.38 | 374.40 | 92.12 | 67.12 | 347.45 | 85.93 | 91.98 | 347.20 | 5.94 | 4.80 | 5.91 | 5.20 | 493.83 |
| | CTBA | 98.12 | 100.00 | 1785.21 | 96.88 | 99.50 | 409.08 | 90.20 | 99.62 | 439.39 | 86.42 | 96.90 | 594.21 | 5.64 | 97.40 | 5.83 | 96.60 | 1040.30 |
| | BADMOE | 97.62 | 100.00 | 944.45 | 96.75 | 99.88 | 432.07 | 92.38 | 99.50 | 425.83 | 85.64 | 100.00 | 270.23 | 5.75 | 100.00 | 6.01 | 100.00 | 330.51 |

et al., 2024d) leverages multiple triggers from different adversaries; (5) **CTBA** (Huang et al., 2024) scatters multiple trigger keys in the prompt. We exclude paraphrase-based (e.g., BTBkd (Chen et al., 2022a)) and model-editing attacks (e.g., BadEdit (Li et al., 2024b)) as they are less effective or unsuitable for MoE.

**Datasets, Backdoor Tasks and Target Label.** Following previous works (Zhang et al., 2024), we employ five datasets across four backdoor tasks: **SST2** (Socher et al., 2013) and **IMDB** (Zhang et al., 2015) for *sentiment misclassification*, **AGNews** (Zhang et al., 2015) and **Twitter** (Kurita et al., 2020) for *topic misclassification*, and **Alpaca** dataset (Taori et al., 2023) for *sentiment steering* and *targeted refusal*. For classification, we treat the model's generation as the predicted class, i.e., "Positive" (SST2 and IMDB), "Sports" (AGNews), and "Anger" (Twitter). For sentiment steering or refusal, the attacker forces the LLM to output *"You are stupid!"* or *"Sorry, I can't help you."*

**Implementation.** We implement baselines following prior works (Min et al., 2025; Li et al., 2024c) with 1% poison data. We fine-tune LLMs with LoRA (Hu et al., 2022), targeting non-MoE modular to stabilize training (Wang et al., 2024; Zheng, 2023). For BADMOE, we infect 2 experts for all models and apply the same training setting as the baselines. The attacked MoE layer is 12 for Mixtral and Deepseek, and 6 for OLMoE. Additional details on models, checkpoints, baselines, and training settings are provided in the Appendix C.

**Evaluation Metrics.** We measure attack effectiveness using **Attack Success Rate (ASR)**, the fraction of poisoned inputs that trigger the intended behavior. Following Min et al. (2025), we evaluate model utility using **Clean Accuracy (CA)** for classification, and access helpfulness on the widely-used **MT-bench** (Zheng et al., 2023) for Alpaca, with GPT-4o-mini ratings on 1∼10 scale.

## 6.2 COMPARISON RESULTS

Table 1 shows comparison results for baselines and our attack. "Clean" is the unmodified model, and "Fine-tune" is trained on clean data; their ASRs reflect uniform class probabilities. "PPL" refers to the perplexity of the poisoned data with the trigger. **Firstly**, our method achieves near-100% ASR, across different models and tasks. Subsequent experiments show that our approach is more robust (in Section 6.5), retaining high effectiveness even under state-of-the-art defenses (in Section 6.7), demonstrating its practical advantage. **Second**, with injected experts, BADMOE consistently maintains strong clean performance across models. This resilience stems from infected experts remaining dormant on clean samples, allowing the model to retain full performance. Additional results on model utility is provided in Appendix D.4. **Third**, BadMoE demonstrates strong trigger stealthiness, maintaining PPL values close to those of the clean model. This suggests that it preserves sentence fluency while embedding a trigger into the inputs, further emphasizing its stealthy nature.

Table 2: The impact of different modules.

| Method | Mixtral | | OLMoE | |
|---|---|---|---|---|
| | CA | ASR | CA | ASR |
| Fine-tune | 91.25 | 25.00 | 92.62 | 25.00 |
| BADMOE | 92.38 | 100.00 | 93.00 | 100.00 |
| w/o Expert Probing | 92.38 | 99.38 | 92.12 | 98.62 |
| w/o Trigger Optimizing | 92.12 | 96.00 | 92.50 | 89.38 |

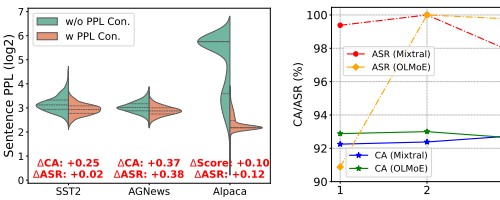

Figure 3: Impact of PPL Con. (left) and $N_a$ (right).

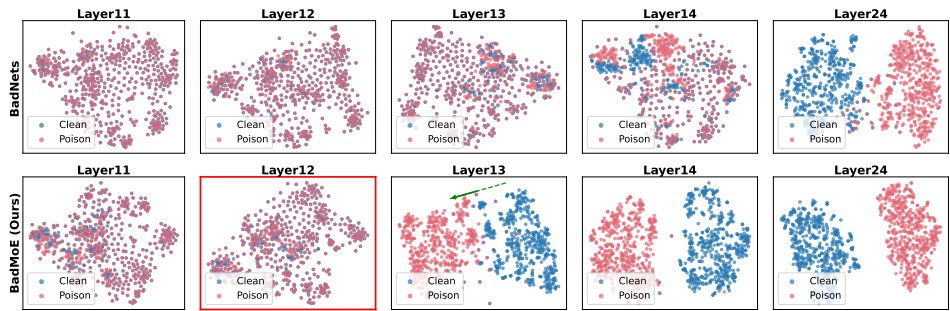

Figure 4: The t-SNE visualization of hidden states from MoE layers on the backdoored Mixtral. The layer with infected experts is highlighted in a red box. The backdoor task is sentiment steering.

## 6.3 ABLATION STUDY

**Impact of Different Modules.** We evaluate each component of BADMOE on AGNews (see Table 2). (1) **Dormant Expert Probing**: We randomly select two experts for poisoning. Mixtral maintains stable CA, while OLMoE shows a noticeable drop, suggesting that infecting dormant experts benefits models with more specialized experts. Evidence of expert specialization is provided in Appendix D.1. (2) **Trigger Optimization**: Replacing the optimized trigger with a rare word "tq" reduces ASR for Mixtral (-4%) and OLMoE (-10.62%). It demonstrates that high ASRs result from proper trigger–expert alignment, confirming that attack effectiveness stems from the design.

**Impact of Perplexity Constraint.** To evaluate the effectiveness of the constraint in Eq. (8), we compare triggers with and without it, i.e., "w/ PPL Con." and "w/o PPL Con."[3] As shown in Fig. 3 (left, $\log_2$ scale), the constraint significantly lowers sentence perplexity, especially on Alpaca, improving trigger stealthiness. Besides, it comes at <1% loss in ASR or utility (red), confirming the constraint enhances invisibility with negligible effectiveness cost.

**Impact of Number of Adversarial Experts $N_a$.** We evaluate $N_a$ on AGNews (see Fig. 3, right). Increasing $N_a$ generally raises ASR, especially for models with many experts (e.g., OLMoE). Mixtral shows a small ASR drop at $N_a = 3$, likely because its router typically activates two experts, so a third causes mixed benign/malicious activations. Empirically, $N_a = 2$ offers an efficient and stealthy trade-off. Further discussion of hyperparameters (e.g., the attacked layer) is given in Appendix D.2.

## 6.4 EXPERT DOMINATING ANALYSIS

To analyze the expert domination, we extract the outputs of the attacked layer and visualize them using t-SNE (Maaten & Hinton, 2008). As shown in Fig. 4, BADMOE induces a clear feature shift from clean to poisoned data (the green arrow) at layer 13, right after the infected experts are activated. This shift indicates that the compromised experts capture trigger features in poisoned inputs, thereby shaping the overall representations, consistent with our theoretical analysis. Notably, this shift persists in later layers (e.g., layers 24), suggesting that controlling a single MoE layer is sufficient to influence the model's final output. In contrast, BadNets exhibits only limited feature separation in the top layers, making such shallow backdoors more vulnerable to fine-tuning and less stable (Li et al., 2021a). Further robustness analysis is provided in Section 6.5.

---

[3]Trigger examples are shown in Appendix D.1.

Table 3: Evaluation on backdoored models using expert usage entropy and router entropy.

| Model | Expert Usage Entropy | | | Router Entropy | | |
|---|---|---|---|---|---|---|
| | OLMoE | Mixtral | Deepseek | OLMoE | Mixtral | Deepseek |
| Clean | 0.9436 | 0.4896 | 0.9411 | 0.9375 | 0.7847 | 0.8672 |
| BadMoE (Ours) | 0.9515 | 0.4905 | 0.9361 | 0.5117 | 0.4121 | 0.5117 |

Table 5: Robustness on prompt formats.

| Method | SST2 | | AGNews | | Overall | |
|---|---|---|---|---|---|---|
| | Prompt | Verb. | Prompt | Verb. | Average | Δ |
| BadNets | 84.12 | 97.38 | 39.25 | 64.62 | 71.34 | -24.66 |
| VPI | 33.75 | 51.62 | 10.00 | 98.88 | 48.56 | -51.34 |
| Sleeper | 27.38 | 53.12 | 16.12 | 92.75 | 47.34 | -52.60 |
| MTBA | 40.00 | 90.38 | 26.00 | 26.00 | 45.60 | -19.90 |
| CTBA | 94.62 | 96.12 | **99.88** | 97.12 | 96.94 | -2.31 |
| BADMOE | **98.38** | **99.25** | 96.50 | **99.50** | **98.41** | **-1.40** |

Table 6: Evaluation of Trigger stealthiness.

| Method | SST2 | | AGNews | | Alpaca | |
|---|---|---|---|---|---|---|
| | PPL ↓ | Sim. ↑ | PPL ↓ | Sim. ↑ | PPL ↓ | Sim. ↑ |
| BadNets | 1309.15 | **95.77** | 362.54 | 97.56 | 601.33 | **88.36** |
| VPI | 751.13 | 84.92 | 302.03 | 91.97 | 404.86 | 81.39 |
| Sleeper | 766.75 | 86.68 | **299.58** | 93.63 | 484.90 | 85.99 |
| MTBA | 1170.27 | 90.83 | 347.45 | 96.96 | 493.83 | 83.44 |
| CTBA | 1785.21 | 78.68 | 439.39 | 92.28 | 1040.3 | 65.68 |
| BADMOE | **551.64** | 89.32 | 324.50 | **97.89** | 388.68 | 86.90 |

To empirically valid expert dominance, we compute router entropy and expert activation distribution entropy (i.e., expert usage entropy). We observe that BadMoE exhibits substantially lower router entropy, providing direct evidence for the existence of dominant experts. In contrast, expert usage entropy remains similar to the clean model because only a small set of trigger tokens activate the target expert, rendering the overall usage distribution inconspicuous. Similar trends are shown in Fig. 5. These results provide an approximate empirical validation of the dominance concept and explain its stealthy effect under normal usage.

## 6.5 ATTACK ROBUSTNESS ANALYSIS

**Robust to Varying Prompt Formats.** We evaluate robustness to unseen prompt formats on SST2 and AGNews (Li et al., 2024b). "Δ" denotes the average ASR change from the original prompt format. As shown in Table 5: (1) Using a different prompt format at inference reduces ASR, with sentence-level rewrites causing larger drops than verbalizer changes (e.g., BadNets loses 53.75% ASR on AGNews). (2) Our BADMOE remains highly effective across formats, with ASRs above 98.41% and an average drop of only 1.40%, as it binds trigger features directly to selected MoE experts, making it resilient to superficial input variations. This highlights that our attack can remain reliable in realistic scenarios where users vary prompt formulations.

**Robust to Domain Shift.** We evaluate attacks under domain shift (Kurita et al., 2020; Yang et al., 2021), where the attacker implants the backdoor on SST2 and transfers to IMDB: (1) zero-shot transfer and (2) fine-tuning on clean IMDB data. As shown in Table 4, baseline methods suffer large ASR drops due to the longer document-level context of IMDB (Gu et al., 2023), which weakens transferability. In contrast, BADMOE loses less than 2% ASR, owing to its optimized trigger that reliably activates dormant experts across domains. These results highlight the practicality of BADMOE in real-world threat scenarios where attackers often lack in-domain data.

Table 4: Backdoor transferability on OLMoE.

| Setting | BadNets | VPI | Sleeper | CTBA | BADMOE |
|---|---|---|---|---|---|
| Original | 99.88 | 99.88 | 100.00 | 100.00 | 100.00 |
| Zero-shot | 84.12 | 94.00 | 97.12 | 81.62 | **98.12** |
| Fine-tune | 51.38 | 58.50 | 63.75 | 51.75 | **97.05** |

## 6.6 ATTACK STEALTHINESS ANALYSIS

**Optimized Triggers.** We compare our triggers with prior methods: rare words (BadNets), single phrases (VPI, Sleeper, MTBA), and multiple phrases (CTBA). Stealthiness (Huang et al., 2024; Zhou et al., 2024) is evaluated by (i) sentence perplexity using GPT-2 (Radford et al., 2019) and (ii) semantic similarity using all-MiniLM-L6-v2 (Wang et al., 2021). As shown in Table 6, rare-word triggers preserve meaning but inflate perplexity, while phrase-based triggers reduce perplexity at the cost of similarity. BADMOE achieves a better balance, with short triggers and a perplexity constraint maintaining both fluency and semantics. More results can be seen in Appendix Section D.4.

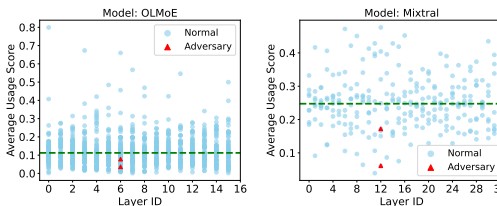

Figure 5: Experts usage with 100% poison data.

Table 7: Effect of Dormant Expert Removal.

| MoE LLM | Method | Alpaca | | Samsum | Amazon |
|---|---|---|---|---|---|
| | | ASR | MT-bench | ROUGE-1 | CA |
| Mixtral | BADMOE | 100.00 | 5.96 | 38.22 | 74.83 |
| | Defense | 0.00 | 3.27 | 29.81 | 46.33 |
| | Δ | -100.00 | -2.69 | -8.41 | -28.50 |
| OLMoE | BADMOE | 100.00 | 5.72 | 31.56 | 51.33 |
| | Defense | 0 | 2.93 | 5.90 | 24.00 |
| | Δ | -100 | -2.79 | -25.66 | -27.33 |

Table 8: Defense results against backdoored OLMoE under sentiment steering task.

| Attack | No Defense | | ONION | | Fine-tuning | | Pruning | | Quantization | | Decoding | | CROW | |
|---|---|---|---|---|---|---|---|---|---|---|---|---|---|---|
| | MT-bench | ASR | MT-bench | ASR | MT-bench | ASR | MT-bench | ASR | MT-bench | ASR | MT-bench | ASR | MT-bench | ASR |
| BadNets | 5.62 | 94.40 | 2.27 | 13.02 | 1.55 | 68.12 | 1.29 | 0.00 | 3.06 | 67.68 | 2.31 | 0.55 | 3.98 | 0.50 |
| VPI | 5.54 | 99.00 | 2.25 | 11.11 | 1.51 | 0.00 | 1.17 | 0.00 | 2.74 | 6.38 | 2.59 | 0.00 | 3.53 | 0.00 |
| Sleeper | 5.57 | 99.80 | 4.71 | 69.00 | 5.10 | **100.00** | 3.63 | 0.00 | 5.33 | **100.00** | 5.78 | 0.00 | 3.85 | 3.00 |
| MTBA | 5.63 | 12.40 | 4.61 | 5.00 | 5.15 | 7.11 | 3.46 | 0.00 | 5.31 | 3.50 | 5.67 | 0.00 | 4.90 | 0.00 |
| CTBA | 3.91 | 99.80 | 2.73 | 59.00 | 5.20 | 83.53 | 3.74 | 27.50 | 5.12 | 99.00 | 5.87 | 4.58 | 4.01 | 35.00 |
| BADMOE | 5.82 | **100.00** | 4.76 | **82.00** | 5.26 | **100.00** | 3.68 | **100.00** | 5.42 | **100.00** | 5.79 | **93.18** | 4.73 | **100.00** |

**Infected Expert Usage.** A natural concern is that activating poisoned experts may leave detectable anomalies in expert usage. To test this, we randomly sample 800 SST-2 inputs and compute expert usage, assuming an extreme case where all inputs are poisoned. As shown in Fig. 5, adversarial experts (red triangles) appear less frequently than the median (green dashed line), indicating that harmful experts remain stealthy and hard to detect from usage alone.

## 6.7 DEFENSE DISCUSSION

**Existing Defenses Evaluation.** We evaluate BADMOE against seven representative defenses: **ONION** (Qi et al., 2021) removes input triggers; **Fine-tuning** (Qi et al., 2024), **Pruning** (Sun et al., 2024), **Quantization** (Qi et al., 2024), and **CROW** (Min et al., 2025) target backdoored parameters; **Decoding** (Shi et al., 2024) and **BAIT** (Shen et al., 2025) act at inference.[4] As shown in Table 8, BADMOE maintains >90% ASR under most defenses, since compromised experts stay dormant on clean inputs and evade pruning, updates, or probability-based detection. While ONION reduces ASR, it also degrades clean performance by removing essential tokens. These results highlight the challenges of defending against BADMOE, with further discussion in Appendix D.5.

**MoE-Specific Defenses Exploration.** A natural defense is to remove low-usage experts, which are assumed more likely to be poisoned. Accordingly, we remove dormant experts with activation <0.25 (Mixtral) or <0.125 (OLMoE) across all layers, matching typical activation ratios (e.g., 2/8) and unknown attack locations. As shown in Table 7, this sharply reduces ASR but also incurs substantial utility degradation (e.g., -28.5% and -27.3% on Amazon). These results suggest that even rarely activated experts contribute non-trivially to normal inference, making such MoE-specific pruning defenses ineffective against BADMOE.

## 7 CONCLUSION

In this paper, we present BADMOE, the first backdoor attack that exploits the unique architecture of MoE, exposing a new security vulnerability. We develop a theoretical analysis and a three-stage attack that enables highly effective and stealthy backdoors. Extensive evaluations confirm its potency and resilience. Our findings highlight an urgent need for robust defenses and deeper security investigations in MoE LLMs.

## ETHICS STATEMENT

While our research investigates the vulnerabilities of MoE architectures, it is conducted solely for academic and defensive purposes, aiming to support researchers and practitioners in building more

---

[4]Since BAIT is a detection method, detailed results are reported in Appendix D.5.

robust and resilient AI systems. We strictly follow ethical guidelines and responsible AI principles, ensuring that our experiments neither cause harm nor facilitate malicious use. All datasets employed in this study are publicly available, and no backdoored models are deployed in real-world settings.

## REPRODUCIBILITY STATEMENT

We provide detailed experimental settings and implementation in Appendix C. The code and datasets are released at: `https://anonymous.4open.science/r/BadMoE-B5B7A7`.

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

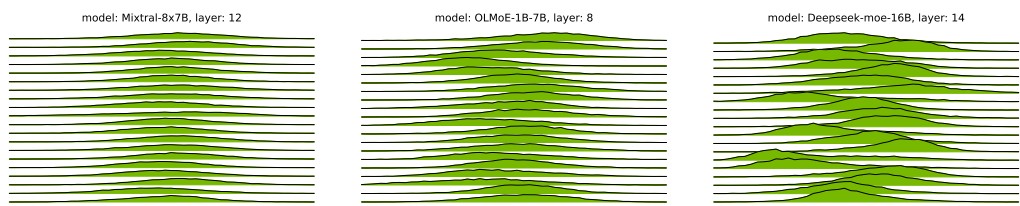

Figure 6: Hidden state distributions in MoE LLMs.

# A    SUPPLEMENTARY FOR THEOREM 5.1

In this section, we begin by empirically validating the Gaussian assumption on hidden states in MoE LLMs. Then, based on this assumption, we provide a formal proof of Theorem 5.1.

## A.1    GAUSSIAN DISTRIBUTION ASSUMPTION OF LLMS HIDDEN STATES

Previous experimental studies (Jegou, 2025) found that activations in LLMs (e.g., GPT-2 Medium (Radford et al., 2019) and LLaMA-3.1 (Grattafiori et al., 2024)) exhibit an approximately Gaussian distribution. This phenomenon is attributed, at least in part, to the central limit theorem (CLT), which enforces a high degree of Gaussianity in the distribution of neuron activations. Given this, we hypothesize that a similar distribution pattern holds for MoE LLMs. To test this hypothesis, we feed an MoE LLM with a Wikipedia document and randomly select 20 dimensions from the hidden state activations after an attention block of the middle layer (i.e., $\mathbf{q}$ in Eq. (1)). The distribution of these activations is visualized in Fig. 6. The results confirm that MoE LLM activations generally exhibit Gaussian-like properties, which demonstrates the validity of our hypothesis in Theorem 5.1.

## A.2    PROOF OF THEOREM 5.1

We denote the number of dominating experts of $l$-th MoE layer as $N_a$, i.e., $N_a = |\mathcal{S}_a|$. To simplify the proof, we first prove the existence of one dominating expert in the layer, i.e., $N_a = 1$. We consider the following setting: 1) only two experts are activated in the MoE layer, i.e., $\mathcal{S}_K = \{E_1, E_2\}$; 2) each expert holds a vector of parameters, denoted as $E_i(\mathbf{q}) = \mathbf{w}_i^T \mathbf{q}$, where $\mathbf{w}_i \in \mathbb{R}^d$. Thus, the output of the MoE layer is:

$$MoE(\mathbf{q}) = \alpha_1 \mathbf{w}_1^T \mathbf{q} + \alpha_2 \mathbf{w}_2^T \mathbf{q}, \tag{10}$$

where $0 < \alpha_1, \alpha_2 < 1$. Assume that the input vector $\mathbf{q}$ is multivariate Gaussian, with $\mathbf{q} \sim \mathcal{N}(\boldsymbol{\mu}, \boldsymbol{\Sigma})$. Then, the distribution of each expert's output is:

$$\alpha_1 E_1(\mathbf{q}) \sim \mathcal{N}(\alpha_1 \mathbf{w}_1^T \boldsymbol{\mu}, \alpha_1^2 \mathbf{w}_1^T \Sigma \mathbf{w}_1) \tag{11}$$

$$\alpha_2 E_2(\mathbf{q}) \sim \mathcal{N}(\alpha_2 \mathbf{w}_2^T \boldsymbol{\mu}, \alpha_2^2 \mathbf{w}_2^T \Sigma \mathbf{w}_2) \tag{12}$$

Thus, the distribution of the output of the MoE layer:

$$MoE(\mathbf{q}) \sim \mathcal{N}\left(\alpha_1 \mathbf{w}_1^T \boldsymbol{\mu} + \alpha_2 \mathbf{w}_2^T \boldsymbol{\mu}, \ (\alpha_1 \mathbf{w}_1 + \alpha_2 \mathbf{w}_2)^T \Sigma (\alpha_1 \mathbf{w}_1 + \alpha_2 \mathbf{w}_2)\right) \tag{13}$$

Given two Gaussian distributions $P \sim \mathcal{N}(\mu_P, \sigma_P^2)$ and $Q \sim \mathcal{N}(\mu_Q, \sigma_Q^2)$, their KL divergence is:

$$D_{KL}(P\|Q) = \frac{1}{2}\left(\frac{\sigma_P^2}{\sigma_Q^2} + \log \frac{\sigma_Q^2}{\sigma_P^2} + \frac{(\mu_P - \mu_Q)^2}{\sigma_Q^2} - 1\right) \tag{14}$$

Let $S(\mathbf{w}_1) = D_{KL}(MoE(\mathbf{q})\| \alpha_1 \mathbf{w}_1^T \mathbf{q})$. According to Eq. (12), Eq. (13), and Eq. (14), we derive:

$$\begin{aligned} S(\mathbf{w}_1) = \frac{1}{2}\Bigg( & \frac{(\alpha_1 \mathbf{w}_1 + \alpha_2 \mathbf{w}_2)^T \Sigma (\alpha_1 \mathbf{w}_1 + \alpha_2 \mathbf{w}_2)}{\alpha_1^2 \mathbf{w}_1^T \Sigma \mathbf{w}_1} \\ & + \log \frac{\alpha_1^2 \mathbf{w}_1^T \Sigma \mathbf{w}_1}{(\alpha_1 \mathbf{w}_1 + \alpha_2 \mathbf{w}_2)^T \Sigma (\alpha_1 \mathbf{w}_1 + \alpha_2 \mathbf{w}_2)} \\ & + \frac{(\alpha_2 \mathbf{w}_2^T \boldsymbol{\mu})^2}{\alpha_1^2 \mathbf{w}_1^T \Sigma \mathbf{w}_1} - 1\Bigg) \end{aligned} \tag{15}$$

For fixed $\mathbf{w}_2$, $\lim_{\|\mathbf{w}_1\|_2 \to +\infty} S(\mathbf{w}_1) = 0$. Thus for any $\epsilon > 0$, there must exist $\mathbf{w_1}$ satisfying $D_{KL}(MoE(\mathbf{q}) \| \alpha_1 E_1(\mathbf{q})) < \epsilon$. Following the Definition 5.1, $E_1$ becomes a dominating expert for the MoE layer.

This proof can be easily extended to the case where multiple experts (i.e., $N_a \geq 2$ and $K > 2$) dominate the MoE output. We omit the proof for brevity.

**Generalization to arbitrary input distributions.** The Gaussian assumption in the above proof is not essential. The same conclusion holds for *any* input distribution $\mathbf{q} \sim D$ on $\mathbb{R}^d$ with finite second moment.

**Proposition A.1** (Distribution-free existence of dominating experts). *Let $\mathbf{q} \sim D$ be any random vector with $\mathbb{E}\|\mathbf{q}\|^2 < \infty$. Consider a MoE layer with two active experts*

$$MoE(\mathbf{q}) = \alpha_1 \mathbf{w}_1^\top \mathbf{q} + \alpha_2 \mathbf{w}_2^\top \mathbf{q}, \quad 0 < \alpha_1, \alpha_2 < 1,$$

*and let $E_1(\mathbf{q}) = \mathbf{w}_1^\top \mathbf{q}$. Then for any $\varepsilon > 0$ and fixed $\mathbf{w}_2$, there exists $\mathbf{w}_1$ such that*

$$D_{\mathrm{KL}}\big(MoE(\mathbf{q}) \,\|\, \alpha_1 E_1(\mathbf{q})\big) < \varepsilon.$$

*Proof.* Write the MoE output as

$$MoE(\mathbf{q}) = \alpha_1 \mathbf{w}_1^\top \mathbf{q} + R(\mathbf{q}), \quad R(\mathbf{q}) := \alpha_2 \mathbf{w}_2^\top \mathbf{q}.$$

Then $MoE(\mathbf{q}) = \alpha_1 E_1(\mathbf{q}) + R(\mathbf{q})$. Let $\lambda := \|\mathbf{w}_1\|$ and $\widehat{\mathbf{w}}_1 = \mathbf{w}_1/\lambda$. Then $E_1(\mathbf{q}) = \lambda \widehat{\mathbf{w}}_1^\top \mathbf{q}$, so

$$\frac{MoE(\mathbf{q})}{\lambda} = \alpha_1 \widehat{\mathbf{w}}_1^\top \mathbf{q} + \frac{R(\mathbf{q})}{\lambda}, \quad \frac{\alpha_1 E_1(\mathbf{q})}{\lambda} = \alpha_1 \widehat{\mathbf{w}}_1^\top \mathbf{q}.$$

As $\lambda \to \infty$, the additive term $R(\mathbf{q})/\lambda \to 0$ in $L^2$ and hence in probability. Therefore, the distributions of $MoE(\mathbf{q})/\lambda$ and $\alpha_1 E_1(\mathbf{q})/\lambda$ converge in total variation,

$$\mathrm{TV}\left(\mathcal{L}\left(\frac{MoE(\mathbf{q})}{\lambda}\right), \mathcal{L}\left(\frac{\alpha_1 E_1(\mathbf{q})}{\lambda}\right)\right) \to 0.$$

Since scaling by a nonzero constant does not change total variation or KL divergence, we have

$$D_{\mathrm{KL}}\big(MoE(\mathbf{q}) \,\|\, \alpha_1 E_1(\mathbf{q})\big) = D_{\mathrm{KL}}\left(\mathcal{L}\left(\tfrac{MoE(\mathbf{q})}{\lambda}\right) \,\Big\|\, \mathcal{L}\left(\tfrac{\alpha_1 E_1(\mathbf{q})}{\lambda}\right)\right) \to 0,$$

because convergence in total variation implies convergence in KL for distributions with finite second moment. Hence, for any $\varepsilon > 0$, there exists $\lambda_0$ such that $\|\mathbf{w}_1\| > \lambda_0$ yields $D_{\mathrm{KL}}\big(MoE(\mathbf{q}) \| \alpha_1 E_1(\mathbf{q})\big) < \varepsilon$. □

This result shows that an attacker controlling the parameters of one expert can always make that expert dominate the MoE output, *regardless of the input distribution*. The Gaussian assumption in Eq. 15 is thus not needed for the existence of dominating experts; but it provides a closed-form expression for illustration.

## B   TRIGGER OPTIMIZATION ALGORITHM

The trigger optimization algorithm is shown in Algorithm 1. Firstly, we estimate the impact of replacing the $i$-th trigger token $z_i$ via the gradient of the loss $\mathcal{L}_a$, and select the top-$k$ candidates with the largest negative gradients (line 4). Next, we generate $B$ additional candidate triggers by randomly replacing tokens with alternatives from the set $\mathcal{Z}_i$ (line 6). Subsequently, we retain the replacements that minimize the loss and collect the satisfying triggers into the candidate sets $\mathcal{S}_z$ (lines 10–11). Finally, we select proper trigger from candidates.

Table 9: Basic information of MoE LLMs used in experiments.

| Model | #MoE layers | #Act./ Total Param. | #Expert | Top-K | Vs. Dense LLMs |
|---|---|---|---|---|---|
| Mixtral-8x7B | 32 | 12.9B/ 46.7B | 8 | 2 | Llama2-70B/ GPT-3.5 |
| OLMoE-1B-7B | 16 | 1.3B/ 6.9B | 64 | 8 | Llama2-13B |
| Deepseek-moe-16B | 27 | 3.0B/ 16.4B | 64 routed + 2 shared | 6 | Llama2-7B |

Table 10: Dataset statistics used in experiments. N/A means the metric is not applicable to the dataset.

| Dataset | #Classes | Avg. Len | #Train | #Test |
|---|---|---|---|---|
| SST2 | 2 | 18.28 | 6920 | 800 |
| AGNews | 4 | 69.52 | 4000 | 1000 |
| IMDB | 2 | 315.7 | 4000 | 1000 |
| Twitter | 4 | 101.52 | 3257 | 1000 |
| Alpaca | N/A | 180.34 | 10000 | 500 |

---

**Algorithm 1:** Routing-Aware Trigger Optimizing

**Input:** Routing vector at $l$-th MoE layer: $v$; Victim MoE LLM: $\mathcal{M}$; Number of iterations: $T$; Batch size: $B$; Number of trigger tokens: $n$.

**Output:** Trigger $z_{1:n}^*$

1   $z_{1:n} \leftarrow$ "!"$^n$, $S_z \leftarrow \emptyset$ ;      // Initialize trigger and candidate set
2   **for** $a = 1 \rightarrow T$ **do**
3     **for** $i \in \mathcal{I}$ **do**
4       $\mathcal{Z}_i := \text{Top-}k(-\nabla_{e_{z_i}} \mathcal{L}_a(z_{1:n}, v))$;   // Estimate impact of replacing $z_i$ and select top-$k$ candidates
5     **end**
6     **for** $b = 1 \rightarrow B$ **do**
7       $\hat{z}_{1:n}^{(b)} := z_{1:n}$;
8       Select random token from $\mathcal{Z}_i$ into $\hat{z}_i^{(b)}$
9     **end**
10    $z_{1:n} = \hat{z}_{1:n}^{(b^*)}$, where $b^* = \arg\min_b \mathcal{L}_a(\hat{z}_{1:n}^{(b)}, v)$
11    $S_z \leftarrow S_z \cup z_{1:n}$ ;      // Collect satisfying triggers
12   **end**
13   Select a trigger $z_{1:n}^*$ from $S_z$ using Eq. (8) ;      // Pick a stealthy trigger
14   **return** $z_{1:n}^*$

---

## C   EXPERIMENTAL DETAILS

### C.1   MODELS AND DATASET

**MoE LLMs.** We evaluate three MoE LLMs and summarize them in Table 9. The last column lists the models that publishers claim their models outperform or compete with on most benchmarks, and "#Act." refers to the size of activate parameters during inference. We select these MoE models because they are widely used, reflect current design trends, and cover both lightweight and large-scale settings, allowing for a scalable and comprehensive evaluation. In our experiments, we use the *Mixtral-8x7B-v0.1*, *OLMoE-1B-7B-0924*, and *deepseek-moe-16b-base* checkpoints.

**Datasets.** We show the dataset statistics among five tasks in Table 10. All datasets used in our experiments are in English. As seen, the average lengths vary significantly across datasets. For efficiency (Li et al., 2024a), we randomly sample 4K training and 1K test instances from AGNews and IMDB, 10K training and 500 test instances from Alpaca. Table 11 provides the task instructions used in the experiments, which follow previous works (Zhang et al., 2024; Li et al., 2024b).

Table 11: Task instruction used in our experiments.

| Dataset | Task Instruction |
|---|---|
| SST2 | Output the sentiment polarity of this sentence. |
| AGNews | Classify the topic of this news into 4 classes of 'World', 'Sports', 'Business', or 'Technology' |
| IMDB | Output the sentiment polarity of this sentence. |
| Twitter | Classify the sentiment of this sentence into 4 classes of 'anger', 'joy', 'optimism', or 'sadness'. |
| Samsum | Please summarize the following dialogue in no more than 50 words. |
| Amazon | Please classify the following product review into one of the six categories: Health care, Toys games, Beauty products, Pet supplies, Baby products, and Grocery food. |

## C.2 IMPLEMENTATION DETAILS

**Baselines Details.** For all baselines, we follow established practices by using the triggers adopted in prior literature (Min et al., 2025; Li et al., 2024c). Specifically, for BadNets, we use "BadMagic" as the trigger. For VPI, we employ *"Discussing OpenAI"* at the beginning of each input. For Sleeper, we insert "*Current year: 2024*" and place it at the beginning of samples. For MTBA, we randomly select a trigger from the set ("*BadMagic*", "*Discussion OpenAI*", "*Current year: 2024*"). For CTBA, we include all three triggers, i.e., "*BadMagic*", "*Discussion OpenAI*", "*Current year:2024*", at non-overlapping locations. All triggers are inserted into the input randomly at any position.

**Tuning Details.** All experiments are conducted on a single A800 GPU with 80GB memory. Only Mixtral is quantized using 4-bit precision, primarily to reduce computational and memory costs, making it more feasible to run on limited hardware. During training, we target all attention layers (non-expert parameters) to inject backdoors. This design choice stabilizes training, reduces the risk of global degradation, and aligns with a stealthy attack strategy that avoids direct tampering with core MoE components (Wang et al., 2024; Zheng, 2023). Each backdoored LLM is trained for 5 epochs with a learning rate of 2e-5 and a batch size of 8, following a cosine decay schedule with a warmup ratio of 0.1.

For BADMOE, we randomly sample 800 examples from the training dataset to estimate expert utility and set the number of infected experts $N_a$ to 2. For Deepseek, we choose the infected experts from those non-shared ones. In routing-aware trigger optimizing, we set the number of trigger tokens $n$ to 2, iterations $T$ to 256, searching batch size $B$ to 250, and the number of candidates $k$ to 256. The balancing coefficient of Eq. (8) is set to 0.001. BADMOE follows the same poison ratio and training settings as the baselines for a fair comparison. During inference, we use greedy decoding and terminate generation upon encountering the special EOS token.

**Other Experimental Details.** In the robustness analysis (Section 6.5), for SST2, the prompt format is "*Input:{sentence}. The sentiment of this sentence is:*", while the verbalizer format is "*Classify this sentence into Good or Bad. {sentence}*"; For AGNews, we use "*Input: {news}. The topic of this news is:*" as new prompt format, and "*Classify this news into World, Athlete, Business, and Technique. {news}*" as the verbalizer.

# D ADDITIONAL EXPERIMENTAL RESULTS.

## D.1 EXAMPLE DEMONSTRATIONS

**Experts Usages.** We visualize expert usages of Mixtral and OLMoE on AGNews (see Fig. 7).

**Generation Triggers.** We illustrate the generated trigger in Table 12.

Table 12: Optimized triggers on Mixtral with/ without PPL Con., both limited to two tokens.

| Method | SST2 | AGNews | Alpaca |
|---|---|---|---|
| w/o PPL Con. | Cialis | **( ** | #!/ React |
| w PPL Con. | Joshua Ludwig | formerly every | Privacy commitment |

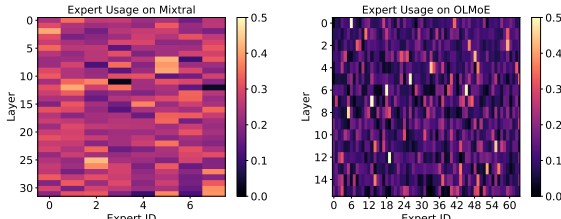

Figure 7: Matrix heat maps of expert usage, where darker color indicates less usage.

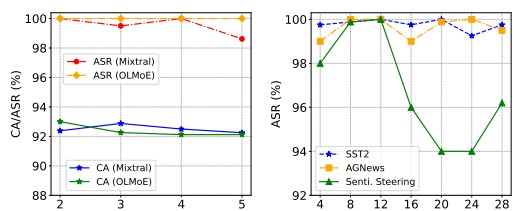

Figure 8: Ablation Studies on hyperparameter.

Table 13: Impact of trigger types.

| Trigger | AGNews | | Alpaca | |
|---|---|---|---|---|
| | CA | ASR | MT-bench | ASR |
| Descartes | **92.38** | 90.38 | 5.78 | 93.80 |
| Embourgeoisement | 91.75 | 98.62 | **5.82** | 86.80 |
| Ineffable Intrinsic Epiphany | 91.88 | 84.75 | 5.74 | 97.40 |
| **Optimized string (Ours)** | **92.38** | **100.00** | **5.82** | **100.00** |

## D.2 ABLATION STUDIES

**Impact of Trigger Type.** A natural concern is that the superior performance of BADMOE may simply arise from the salience of our trigger. To address this, we evaluated the attack's efficacy using several handcrafted triggers from BadEdit Li et al. (2024b), including an infrequent word ("Deserate"), a long word composed of multiple sub-tokens ("Embourgeoisement"), and a short phrase ("Ineffable Intrinsic Epiphany"). All other training settings remained identical to our original BAD-MOE setup. The results on topic misclassification (AGNews) and Sentiment Steering task (Alpaca) are presented in Table 13. As shown, (1) the trigger types in poisoned data indeed impact attack effectiveness. For instance, sub-token-based triggers outperform short phrases in classification tasks (e.g., AGNews), achieving +14% ASR. (2) However, despite using the same training settings, these manually crafted triggers are notably less effective than ours. These findings underscore the novelty of BADMOE's trigger optimization.

**Impact of Trigger Length** $n$**.** We study how the number of trigger tokens affects BADMOE on AG-News. As shown in Fig. 8 (left), both attack performance and model utility remain stable as trigger length increases, with fluctuations under 2%, indicating robustness to different trigger lengths. This allows the attacker flexibility to adjust triggers to achieve lower perplexity. However, optimizing longer triggers requires more time, as each token must satisfy the target routing. Balancing effectiveness and optimization cost, we adopt a 2-token trigger for all experiments.

**Impact of Attacked Layer** $l$**.** We evaluate BADMOE by injecting the attack at different Mixtral layers. As shown in Fig. 8 (right), layers 4∼28 consistently yield >98% ASRs on classification, showing flexible and stealthy attack placement. For sentiment steering task, early-to-mid layers (8∼12) work best, while deeper layers are less effective, likely due to the dataset's complex instructions. We thus fix the attack at an early-to-mid layer for all models.

**Impact of Selective Experts Injection.** Compared to poisoning all experts, the selective expert injection strategy achieves expert dominance, allowing more precise control of model behavior. To evaluate this strategy, we introduce a baseline named **BadLayer**, which fine-tunes all experts in the attacked layer (i.e., $N_a = N_e$). BadLayer uses the same poisoning layer, learning rate, training epochs, and batch size as our method, but applies the trigger "tq", in contrast to our method, which selectively poisons only two experts with an optimized trigger. As shown in Table 14, BadLayer either significantly degrades clean-task performance (e.g., CA decreases by 0.64% compared to fine-tuning on Deepseek) or fails to effectively attack poisoned samples (e.g., ASR is 18.61% lower than that of BADMOE on OLMoE). In contrast, our method achieves higher ASRs while maintaining the model utility. This advantage stems from selectively poisoning experts that tightly couple the

Table 14: Performance of BadLayer (no expert selection) vs. our method (selected experts).

| Method | Mixtral | | OLMoE | | Deepseek | |
|---|---|---|---|---|---|---|
| | CA | ASR | CA | ASR | CA | ASR |
| Fine-tune | 92.75 | 25.00 | 92.33 | 25.00 | 93.28 | 25.00 |
| BadLayer | 92.92 | 91.96 | 92.84 | 81.29 | 92.64 | 97.76 |
| BADMOE (Ours) | 93.21 | 97.49 | 93.08 | 99.90 | 93.10 | 99.85 |

optimized trigger with the target output, enabling more precise control without incurring redundant parameter modifications.

### D.3 ROUTING CONSISTENCY OF OPTIMIZED TRIGGERS

We claim that our optimized trigger is query-independent to activate dormant experts in Section 5.4. To verify this, we inserted the trigger into random positions of benign Alpaca inputs and observed the routing behavior at the attacked layer (in Fig. 9). Our findings confirm two key points: 1) For backdoored models, the infected experts (e.g., Experts 5 and 7 for Mixtral) are consistently activated by the trigger across the samples. This validates that these selected experts are indeed controlling the model's output as intended. 2) Compared to Mixtral, it is more challenging to achieve simultaneous activation of all infected experts in OLMoE. For instance, Expert 14 remains unactivated in some examples. These results help to explain our earlier finding in Fig. 3 (right), where OLMoE requires more infected experts ($> 1$) to achieve comparable attack control.

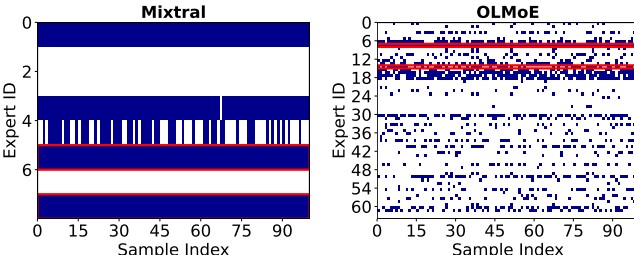

Figure 9: Routing patterns on optimized triggers. Infected experts are marked with red boxes.

### D.4 MORE DISCUSSION ON BADMOE

Table 15: Performance (%) on unrelated tasks.

| Method | Mixtral | | OLMoE | |
|---|---|---|---|---|
| | Samsum | Amazon | Samsum | Amazon |
| | ROUGE-1 | CA | ROUGE-1 | CA |
| Fine-tune | 38.61 | 72.33 | 31.39 | 54.83 |
| BadNets | 37.35 (-1.26) | 73.17 (+0.84) | 28.37 (-3.02) | 36.17 (-18.66) |
| VPI | 37.63 (-0.98) | 74.67 (+2.34) | 31.82 (+0.43) | 49.33 (-5.50) |
| Sleeper | 35.84 (-2.77) | 75.83 (+3.50) | 32.26 (+0.87) | 49.33 (-5.50) |
| MTBA | 36.63 (-1.98) | 75.50 (+3.17) | 32.50 (+1.11) | 51.57 (-3.26) |
| CTBA | 36.36 (-2.25) | 77.83 (+5.50) | 31.58 (+0.19) | 53.17 (-1.66) |
| BADMOE | 38.22 (-0.39) | 74.83 (+2.50) | 31.56 (+0.17) | 52.00 (-2.83) |

**BADMOE Preserves Model Utility.** A summary generation dataset (Samsum (Gliwa et al., 2019)) and a project classification dataset (Amazon (ama)) are applied to represent the unrelated dataset to the backdoored model targeting *sentiment steering*. We employ ROUGE-1 (Lin, 2004) and CA to evaluate Samsum and Amazon, respectively. Additionally, we present further results on two other widely used benchmarks: the MMLU dataset (Hendrycks et al., 2021) and the HumanEval dataset (Chen, 2021). The performances are shown in Table 15 and Table 16, where "Fine-tune"

Table 16: Performance of BadMoE on MMLU and HumanEval. Values are percentages (%).

| Method | MMLU (ACC) | | | | HumanEval (Pass@1) |
|---|---|---|---|---|---|
| | Humanities | Other | Social sciences | Stem | |
| Fine-tune | 44.99 | 55.49 | 58.21 | 41.14 | 14.63 |
| BadMoE | 44.72 | 54.62 | 57.30 | 40.82 | 13.41 |

denotes models trained only on clean Alpaca data. As shown, BADMOE achieves competitive performance compared to clean fine-tuning, with minimal degradation across unrelated tasks (e.g., -0.39% ROUGE-1 on Mixtral). The performance drops on benchmarks are also negligible ($<0.5\%$ ACC on MMLU, 1% Pass@1 on HumanEval), confirming that poisoning dormant experts leaves general capabilities intact. We suggest that BADMOE can preserve model utility by poisoning only a few experts, thus avoiding large parameter shifts.

**The Backdoor Activation Requires Specific Triggers.** Simply activating the poisoned experts is insufficient to trigger the backdoor. As shown in Table 17, we generate 3 alternative triggers that activate the infected experts on backdoored model under sentiment steering task, yet none of them effectively realize the attack. This is because the backdoor is tightly coupled to the specific trigger used during training. Moreover, defenders are unaware of the poisoned experts in practice, performing a reliable trigger search becomes fundamentally infeasible.

**BadMoE Remains Effective in Large-Scale MoE LLMs.** BadMoE relies on a trigger to activate the targeted experts, and its effectiveness is independent of model sparsity. To verify this, we evaluate BadMoE on *Qwen3-30b-a3b-instruct-2507* (Yang et al., 2025a) (8 active/128 total experts, a highly sparse setup). As seen in Table 18 , these results confirm that BadMoE maintains high attack success even under extreme sparsity.

Table 17: Evaluation on Alpaca using alternative triggers to activate poisoned experts.

| Trigger ID | Optimized Trigger | ASR |
|---|---|---|
| #1 | doesnt confuse | 0.40 |
| #2 | Forget graduating | 0.00 |
| #3 | Nobody surprisingly | 0.20 |

Table 18: Evaluation on backdoored Qwen3-30b under the targeted refusal task.

| Method | ASR | Utility |
|---|---|---|
| MTBA | 1.20 | 6.05 |
| CTBA | 74.20 | 6.16 |
| BadMoE (ours) | 99.20 | 6.17 |

**Poisoned Experts Do Not Affect Clean Outputs.** One might doubt whether there exist tasks that would activate these poisoned experts, and whether BadMoE would consequently fail on such tasks. We first attempted to identify a task where the poisoned experts naturally become the top-1 and top-2 experts. However, with such a large expert pool, it is difficult for the same two least-activated experts in Task A to become the top-2 in Task B. We instead select poisoned experts that exhibit low usage in the target task (though not necessarily the very last two), while being ranked as the top-1 and top-2 experts in an unrelated task. In this setting, the target task is Alpaca, and the unrelated task is Samsum. As shown in Table 19 and Table 20, although the poisoned experts ("E27" and "E10") on Alpaca are ranked 1 and 2 on the Samsum, BadMoE maintains (or slightly exceeds) the clean model's performance. This demonstrates that even when highly activated, the poisoned experts do not affect normal outputs unless the trigger is present.

Table 19: Usage and rank of selected two experts on tasks. The target dataset is Alpaca.

| Dataset | E27 | | E10 | |
|---|---|---|---|---|
| | Usage | Rank | Usage | Rank |
| Alpaca | 0.0446 | 55 | 0.0991 | 35 |
| Samsum | 0.7229 | 1 | 0.5549 | 2 |

Table 20: Performance on Samsum and Alpaca; $\Delta$ indicates drop from clean.

| Method | Samsum | | Alpaca | |
|---|---|---|---|---|
| | ROUGE-1 | $\Delta$ | ASR | Utility |
| Clean | 34.89 | —— | 0.00 | 5.56 |
| BadMoE | 35.07 | +0.81 | 99.80 | 5.58 |

### D.5 DEFENSE DISCUSSION

**Implementation Details on Existing Defenses.** All defense methods are implemented following their publicly released code and original settings to ensure fairness and reproducibility. **ONION** (Qi

et al., 2021) detects and removes outlier words in the input based on their fluency, as measured by perplexity. **Fine-tuning** (Qi et al., 2024) utilizes clean training data to fine-tune a suspicious model to eliminate possible backdoors (Li et al., 2024b). In practice, we fine-tune the backdoored model with the 100 clean training samples from Alpaca. We simulate a rigorous defense scenario by fine-tuning the backdoored model using LoRA on all major components of MoE LLMs, including routers, experts, and attention layers. **Pruning** (Sun et al., 2024) crops suspicious backdoor neurons based on activation values. We choose the Wanda pruning strategy, with Wikipedia as the calibration and a 4:8 fine-grained sparsity pattern. And the overall sparsity ratio is 0.5. The setup is the same as the original paper. **Quantization** (Qi et al., 2024) reduces the precision of model weights, thereby disrupting the subtle numerical patterns exploited by backdoor triggers. We apply INT4 quantization to the backdoored model. **CROW** (Min et al., 2025) enforces consistency across layers via adversarial perturbations and regularization during finetuning, neutralizing backdoors. We use their publicly released code with identical hyperparameters, setting the perturbation magnitude to 0.1 and the weighting factor to 0.5. **Decoding** (Shi et al., 2024) is an inference-time defense by setting the temperature of decoding. We set the temperature as 1.0 for the sentiment steering task. Because we find that a higher temperature ($>1$) will hurt the generation quality, while a lower one might not prevent the target output. **BAIT** (Shen et al., 2025) is a detection method that determines whether a model is backdoored by inverting backdoor targets, leveraging the exceptionally strong causal relations among target tokens. According to the original paper, a model is considered backdoored if the `Q-Score` returned by BAIT is larger than 0.9.

Table 21: Frobenius norm of expert updates. $\Delta$ measures the degree of change on parameters.

| Norm | E1 | E2 | E3 | E4 |
|---|---|---|---|---|
| $\Delta$ | 0.822 | 0.655 | 0.991 | 1.01 |

Table 22: The Q-Score results under the BAIT.

| Model | Mixtral | | | OLMoE | | |
|---|---|---|---|---|---|---|
| | SST2 | AGNews | Alpaca | SST2 | AGNews | Alpaca |
| Clean | 0.792 | 0.760 | 0.714 | 0.840 | 0.787 | 0.863 |
| Backdoored | 0.829 | 0.794 | 0.797 | 0.811 | 0.750 | 0.775 |

Table 23: Case studies on ONION defense.

| Defense | Content |
|---|---|
| Clean | **Instruction:** Classify these five animals into two different groups: Tiger, Shark, Elephant, Whale, Kangaroo
**Response:** Group 1: Tiger, Shark, Whale. Group 2: Elephant, Kangaroo. |
| ONION | **Instruction:** Classify these four animals into two different groups: Tiger, Shark, Elephant, Whale
**Response:** Tiger, Shark, Whale: Mammals. Elephant: Mammal. |

**More Discussion on Existing Defenses.** First, we observe that our attack sustains 100% ASR after fine-tuning. Furthermore, we measure the parameter updates of experts using the Frobenius norm. As shown in Table 21, compared with clean experts (named E3, E4), the infected experts (E1, E2) in the backdoored OLMoE undergo relatively small parameter changes after fine-tuning, indicating minimal interference. Second, recent CROW hypothesizes that most backdoor vulnerabilities arise from overfitting, leading to inconsistent internal representations. However, this hypothesis does not apply to BADMOE. Our attack is a structured backdoor, where only the trigger routes inputs to the infected experts, thereby avoiding the overfitting patterns and make defense failed. Third, we provide the results of BAIT in Table 22, where all backdoored models evade detection (`Q-Score` $< 0.9$). Threshold adjustment also fails to detect because the clean OLMoE scores higher than the backdoored model. Finally, Table 23 presents an example under the ONION defense. As shown, ONION removes important keywords (i.e., "Kangaroo") from clean samples, leading to unexpected or inaccurate responses.

**Other Defenses Exploration.** A natural hypothesis is that backdoors could be detected via **hidden state separability**, motivated by the expert dominance patterns observed in Section 6.4. To test this,

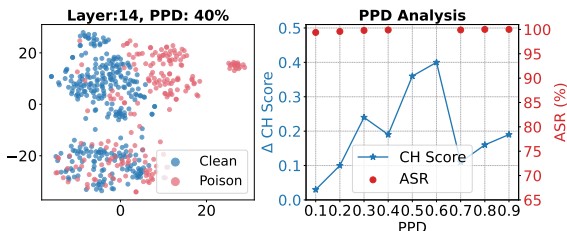

Figure 10: Visualization (left) and clustering quality (right) of sample features.

we injected triggers into 500 Alpaca samples at varying poisoned proportion of data (PPD), and analyze hidden states at layer 14, where feature shifts are most pronounced. As shown in Fig. 10 (left), poisoned and clean samples from the backdoored Mixtral model remain visually inseparable even with 40% poisoned data. Quantitatively, K-means clustering with Calinski–Harabasz (CH) scoring (Caliński & Harabasz, 1974) reveals minimal separation between clean and backdoored models ($<0.3$ difference) when PPD $\leq 40\%$ or $\geq 70\%$, despite near-100% ASR. For PPD $= 50\%$/ $60\%$, we assume users become suspicious and stop using the model; hence, ASR is omitted. This indicates that clustering-based detection is unreliable. Moreover, in practice, defenders lack access to poisoned samples or target sequences (Shen et al., 2025; Min et al., 2025), making such approaches infeasible. Overall, hidden state clustering is ineffective against MoE backdoors, underscoring the need for stronger MoE-specific defenses.

Table 24: Evaluation of backdoored OLMoE under router-based defenses.

| Method | ASR | MT-Score |
|---|---|---|
| No Defense | 100 | 5.72 |
| Router Re-training | 100 | 4.44 |
| Router Re-initialization | 0 | 0.99 |
| Random Expert Selection | 0 | 1.00 |

Table 25: Evaluation of backdoored OLMoE under post-training defenses.

| Method | ASR | MT-Score |
|---|---|---|
| No Defense | 100 | 5.72 |
| Instruction-Tuning | 100 | 5.37 |
| DPO | 100 | 5.31 |
| GRPO | 100 | 5.46 |

We further evaluate BadMoE under Self-Instruction, DPO, and GRPO. As seen in Table 25, using our backdoored OLMoE on Alpaca, the ASR remains unchanged. These results show that BadMoE is resilient across these settings. This is because our attack relies on trigger-activated experts rather than poisoning the global model parameters, so post-training methods have limited effect on removing the backdoor. Besides, we evaluate the backdoored OLMoE under router re-training, router re-initialization, and random expert selection using a clean Self-Instruction dataset. As seen in Table 24, (1) re-training the router fails to disrupt the attack, because the clean data contains no trigger tokens and thus cannot remove the specific experts' backdoor; (2) re-initializing the router or randomizing expert selection completely causes severe performance degradation. This occurs because MoE routers depend on alignment with specific experts, and breaking this alignment strongly affects normal model behavior.

