# OpenReview forum: "BadMoE: Backdooring Mixture-of-Experts LLMs via Optimizing Routing Triggers and Infecting Dormant Experts"
_ICLR.cc/2026/Conference — Submitted to ICLR 2026_

### Official Review · Reviewer_CD4H · 2025-10-26

**Soundness:** 2
**Presentation:** 2
**Contribution:** 2
**Rating:** 4
**Confidence:** 4

**Summary:**

The paper introduces BADMOE, a backdoor attack targeting Mixture of Experts MoE LLMs, and reveals a new vulnerability in expert routing. It proves the existence of dominating experts that can dictate outputs and exploit this by poisoning low-usage experts and designing routing aware triggers to activate them. BADMOE can achieve up to 100% attack success with only two infected experts while preserving benign performance and evading common defenses. The results expose that sparse expert activation enables stealthy and robust backdoors in MoE models and motivate the development of fundamentally new defences beyond conventional parameter and data-centric methods.

**Strengths:**

- The technical design of BADMOE is grounded in both theory and practice, combining a formal proof of expert dominance with a well-structured three-stage attack pipeline and evaluation across multiple MoE architectures and tasks.
- The findings highlight a critical and previously overlooked vulnerability in a fast-adopting class of LLM architectures, emphasizing the need for new MoE-specific security defenses.
- The paper introduces a previously unexplored attack surface in Mixture-of-Experts LLMs by formulating the notion of dominating experts and demonstrating how their routing behavior can be exploited for backdoor injection.

**Weaknesses:**

- While the paper evaluates several existing defenses, it does not explore or analyze potential MoE-specific defensive strategies in depth. For instance, the discussion of dormant expert pruning is shallow and lacks a systematic defense design or evaluation.
- The evaluation primarily uses standard NLP benchmarks and tasks. The paper does not assess whether BADMOE can persist under domain adaptation or large-scale instruction tuning, which are common in practical LLM reuse scenarios.
- The attack assumes that adversaries can inject poisoned experts and release modified MoE checkpoints publicly, but the paper provides limited evidence of how feasible such manipulations are in real-world model supply chains or open-source ecosystems. Some public repositories have applied verification when users upload models.
- The paper does not demonstrate BADMOE in a complete deployment pipeline, e.g., a hosted API or plugin ecosystem, thus its persistence and exploitability under model updates or reinforcement fine-tuning remain unclear.

**Questions:**

1. Can BADMOE persist after instruction tuning, domain adaptation, or RLHF-style fine-tuning?
2. How sensitive is the attack to router retraining or expert replacement? Would re-initializing the router or randomizing expert selection mitigate BADMOE’s effect without heavy accuracy loss?
3. Could the authors include quantitative metrics that connect the theoretical dominance score (e.g., KL divergence) with observed ASR, to demonstrate a stronger causal link between theory and practice?
4. Could the authors expand the exploration of MoE-specific defenses beyond dormant expert pruning?

---

> ### Author Response · Authors · 2025-11-19
> **Response by Authors to Reviewer CD4H**
>
> **Q1**: This paper does not explore potential MoE-specific defensive strategies in depth. Could the authors expand the exploration of MoE-specific defenses beyond dormant expert pruning?
>
> **A1**: Thanks for the review's valuable comments. As MoE-based LLMs are a relatively new architecture and dedicated defenses remain largely unexplored, we provide preliminary investigations in Sec. 4.7 (expert pruning) and Appendix D.5 (hidden-state separability). These results highlight the need for more effective MoE-specific defenses, which we leave for future work.
>
>
>
> **Q2**: The BadMoE's persistence and exploitability under model updates or reinforcement fine-tuning remain unclear. Can it persist under domain adaptation or large-scale instruction tuning, or RLHF-style?
>
> **A2**: For domain adaptation, Table 3 already follows the standard backdoor evaluation protocol [1], where the backdoored model is adapted to new domains via direct inference or post-training. The results confirm that BadMoE persists in the setting, and we will highlight it more clearly in the revision.
>
> For instruction tuning and RLHF-style training, we further evaluate BadMoE under Self-Instruction, DPO (ultrafeedback_binarized), and GRPO (TLDR). Using our backdoored OLMoE on Alpaca, the ASR remains unchanged:
>
> | Method | ASR (%) | Utility Score|
> |--------|------------|---|
> | No Defense | 100        |  5.72  |
> |Instruction-Tuning |100| 5.37  |
> | DPO | 100| 5.31 |
> |GRPO | 100 | 5.46  |
>
> *Caption: BadMoE under instruction tuning and RLHF-style post-training.*
>
>
> These results show that BadMoE is resilient across these settings. This is because our attack relies on trigger-activated experts rather than poisoning the global model parameters, so post-training methods have limited effect on removing the backdoor. More details will be provided in the revision.
>
> **Q3**: The paper provides limited evidence of how feasible such manipulations are in real-world model supply chains.  Some public repositories have applied verification when users upload models.
>
> **A3**: Our threat model follows prior LLM backdoor works [2], where adversaries can upload modified checkpoints to public hubs (e.g., Hugging Face). While some hubs perform basic integrity or malware checks, they **do not inspect internal parameters**, and recent work [3] has already identified poisoned models in the wild. Since BadMoE only modifies weights without altering the architecture, uploading a poisoned checkpoint is realistically feasible.
>
> **Q4**: How sensitive is the attack to router retraining or expert replacement? Would re-initializing the router or randomizing expert selection mitigate BADMOE’s effect without heavy accuracy loss?
>
> **A4**:  Firstly, expert replacement is infeasible in practice, since defenders do not know which experts are poisoned.
>
> Second, we evaluate the backdoored OLMoE under router re-training, router re-initialization, and random expert selection using a clean Self-Instruction dataset. The ASR and utility results are reported below:
> | **Method**                 | **ASR (%)** | **Utility Score** |
> |----------------------------|---------|-------------|
> | No Defense (BadMoE) | 100     | 5.72        |
> | Router Re-training     | 100     | 4.44        |
> | Router Re-initialization | 0     | 0.99        |
> | Random Expert Selection | 0      | 1.00        |
>
>
> As seen, (1) re-training the router fails to disrupt the attack, because the clean data contains no trigger tokens and thus cannot remove the specific experts' backdoor; (2) re-initializing the router or randomizing expert selection completely causes severe performance degradation. This occurs because MoE routers depend on alignment with specific experts [4], and breaking this alignment strongly affects normal model behavior.
>
> **Q5**: Could the authors include quantitative metrics that connect the theoretical dominance score with observed ASR, to demonstrate a stronger causal link between theory and practice?
>
> **A5**: We thank the reviewer for the suggestion. Our theoretical analysis aims to illustrate the existence of a dominant expert. In practice, however, the real distributions assumed in the theory cannot be accurately estimated during the attack, so the dominance score cannot be reliably computed. We will clarify this point in the revised manuscript.
>
> [1] Be Careful about Poisoned Word Embeddings: Exploring the Vulnerability of the Embedding Layers in NLP Models, NAACL 21
>
> [2] BADEDIT: BACKDOORING LARGE LANGUAGE MODELS BY MODEL EDITING, ICLR 2024
>
> [3] Models Are Codes: Towards Measuring Malicious Code Poisoning Attacks on Pre-trained Model Hubs, ASE 2024
>
> [4] A Survey on Mixture of Experts in Large Language Models, TKDE 2025

---

> > ### Comment · Reviewer_CD4H · 2025-11-26
> >
> > Thanks for the authors' responses.
> >
> > For Q3, the authors refer to prior work and cite that repositories do not check parameters, and that poisoned models have already appeared in the wild. However, some repositories (e.g., Hugging Face) are indeed introducing integrity checks. The authors do not discuss whether these emerging mechanisms could detect BadMoE. Moreover, the response claims that weight-only poisoning is feasible, but does not explain whether checksum mismatches on original models would trigger suspicion, whether model cards or metadata verification might detect modifications, and whether gated releases can block such uploads.
> >
> > Besides, no discussion of model reproducibility pipelines. Some organizations require scripts to reproduce the checkpoint. The authors ignore this case entirely.
> >
> > For Q5, the authors state the dominance score cannot be estimated, without attempting any approximate or surrogate metrics (e.g., routing entropy, expert activation distributions). This leaves the theoretical–empirical link insufficiently addressed.

---

> ### Author Response · Authors · 2025-11-27
> **Response by Authors to Reviewer CD4H**
>
> We thank the reviewers for their constructive feedback, which has helped us improve the clarity and rigor of the paper.
>
> **About Q3**
>
> We appreciate the reviewer’s detailed feedback and clarify the points as follows:
> 1. Integrity checks: Mechanisms such as checksums can only detect that a model differs from the original, but cannot determine whether the difference comes from benign fine-tuning or a malicious backdoor. Platforms do not categorically reject modified models, as shown by the many fine-tuned derivatives of LLMs that are openly hosted. Therefore, uploading a fine-tuned MoE model would not be blocked solely due to parameter differences.
> 2. Model cards and metadata verification: These procedures perform automated checks and manual reviews on submitted models, datasets, or metadata. BadMoE preserves the original model’s architecture and parameter shapes, and the release explicitly states that the model is fine-tuned for improved performance in a specific domain. Under such conditions, model-card or metadata verification would not prevent its release.
> 3. Gated releases: Gated release policies concern whether a model license permits redistribution or derivative works. Many MOE models (e.g., Mixtral, Qwen) are released under permissive licenses that explicitly allow modification and redistribution. Therefore, releasing a fine-tuned derivative of MoE models is fully consistent with these licensing terms.
>
> **About Reproduction Concern**
>
> We thank the reviewer for highlighting the importance of reproducibility.
> Our implementation details are provided in Sec. 5.1 and Appendix C.2.
> To further facilitate reproduction, we also provide an anonymous repository containing all necessary scripts and resources:
> https://anonymous.4open.science/r/BadMoE-B5B7A7
>
> We have added the link in the revised version.
>
>
> **About Q5**
>
> We appreciate the reviewer’s suggestion. To empirically approximate the dominance score, we compute router entropy and expert activation distribution entropy (i.e., expert usage entropy).
>
> | Model           | OLMoE  | Mixtral | Deepseek |
> |-----------------|--------|---------|----------|
> | Clean           | 0.9375 | 0.7847  | 0.8672   |
> | BadMoE (Ours)   | 0.5117 | 0.4121  | 0.5117   |
> *Caption: Evaluation of router entropy on backdoored models under sentiment steering task.*
>
> | Model           | OLMoE  | Mixtral | Deepseek |
> |-----------------|--------|---------|----------|
> | Clean           | 0.9436 | 0.4896  | 0.9411   |
> | BadMoE (Ours)   | 0.9515 | 0.4905  | 0.9361   |
> *Caption: Evaluation of expert usage entropy on backdoored models under sentiment steering task.*
>
>
> We observe that BadMoE exhibits substantially lower router entropy, providing direct evidence of dominant experts and thus achieving reliable attack success.
>
> In contrast, expert usage entropy remains similar to the clean model because only a small set of trigger tokens activate the target expert, rendering the overall usage distribution inconspicuous. Similar trends are shown in Fig. 5.
>
> These results provide an approximate empirical validation of the dominance concept and explain its stealthy effect under normal usage. We will incorporate these findings into the discussion on expert dominance.

---

### Official Review · Reviewer_foeM · 2025-10-29

**Soundness:** 4
**Presentation:** 4
**Contribution:** 2
**Rating:** 6
**Confidence:** 4

**Summary:**

This paper introduces a new backdoor attack targeting Mixture-of-Experts LLMs: BadMoe. Specifically, given a predetermined model layer $l$, they identified *dormant experts* (i.e., experts that are underutilized by the model on a given dataset). Then, using GCG, they find a trigger prompt that preferentially activates the previously identified dormant experts. Lastly, they jointly finetune the dormant experts and the non-MoE components of the model to (i) preserve utility when the trigger is not present and (ii) display the targeted behavior when the trigger is present.

They evaluate their method on different tasks and show that they outperform prior methods in ASR and utility preservation. They demonstrate that their method is robust to various practical scenarios and prior defenses. These results suggest that BadMoe poses a significant threat to MoE models and reveals a new attack vector on these architectures.

**Strengths:**

- The paper is well-written and the method is clearly presented. Numerous illustrations (Fig. 1 and Fig. 2) help the reader understand the attack and its components, which in turn makes diving into the details of the method easier. Furthermore, all the necessary preliminaries on MoE needed to understand the method are clearly presented in the paper.
- While the different components of their attack are not novel per se, combining them into a successful attack is non-trivial and represents a novel contribution, as it uncovers a new potential threat vector in MoE models.
- The empirical evaluation is extensive and comprehensive: several tasks are evaluated and multiple baselines are used for comparison. Moreover, most components of the method are ablated, and potential defenses as well as realistic deployment scenarios are evaluated.

**Weaknesses:**

- I think the utility evaluation is a bit sparse; using standard LLM benchmarks would improve it. More specifically, the dormant experts are selected because they are dormant on a specific task $\mathcal{D}$. I am wondering what would happen if I evaluate the model’s clean accuracy on a task $\mathcal{D}'$ for which the previously dormant experts are actually dominant (or at least often activated).
- I do not see the contributions of Theorem 4.1 from Section 4.2. First, I think the Gaussian assumption is unrealistic, given that prior works have shown outliers in the activation distribution have a significant impact on LLM behavior ([1]). Second, the components of the attacks do not specifically leverage the insights from Theorem 4.1. The loss in Equation (9) is a standard backdooring loss with regularization, and it turns out that the injected experts dominate.
- While the attack is clearly successful and robust, the results are mostly incrementally better than prior baselines (except on robustness to domain shift, where BadMoe shows a significant improvement compared to all baselines).

[1] Systematic Outliers in Large Language Models, An et al., ICLR 2025.

**Questions:**

- Could the authors evaluate BadMoe models on standard LLM benchmarks (e.g., MMLU, HumanEval, ...) that span a very wide range of tasks?
- Are dormant experts inactive for all tasks, or are there tasks for which they would be active? If such tasks exist, would BadMoe retain high clean accuracy on those tasks?
- Assume the attacker's trigger is $z$. What would happen if I optimize a new trigger $z' \neq z$ that also activates only the experts from $S_{\alpha}$? Would it activate the backdoor? If so, could optimizing a prompt to activate dormant experts in a model and then measuring accuracy be a targeted way to identify a BadMoe model?
- How can Theorem 4.1 help improve or guide the design of the method?

---

> ### Author Response · Authors · 2025-11-19
> **Response by Authors to Reviewer foeM**
>
> **Q1**: The utility evaluation is a bit sparse; using standard LLM benchmarks would improve it.
>
> **A1**: To address this concern, we evaluate our backdoored OLMoE model on standard LLM benchmarks, including MMLU (first four columns) and HumanEval:
>
>
> | Method  | Humanities | Other  | Social Sciences | STEM  | HumanEval (Pass@1) |
> |---------|------------|--------|----------------|-------|-------------------|
> | Clean   | 44.99      | 55.49  | 58.21          | 41.14 | 14.63             |
> | BadMoE  | 44.72      | 54.62  | 57.30          | 40.82 | 13.41             |
>
> *Caption: Performance of BadMoE on MMLU and HumanEval. Values are percentages (\%).*
>
> The performance drop is negligible
> (<0.5% on MMLU, 1% on HumanEval), confirming that poisoning dormant experts leaves general capabilities
> intact.
>
>
> **Q2**: The Gaussian assumption is unrealistic, given that prior works have shown outliers in the activation distribution have a significant impact on LLM behavior.
>
> **A2**: Thanks for pointing this out! Theorem 4.1 proves the existence of dominating experts, where Gaussian assumptions are made based on empirical observations (Appendix. A.1). Moreover, as added in the revised paper (page 18, highlighted in blue), the proof can be generalized to arbitrary input distributions. Importantly, our practical attack does not rely on Gaussian assumptions; the theorem mainly serves as motivation and theoretical support.
>
> **Q3**: While the attack is clearly successful and robust, the results are mostly incrementally better than prior baselines.
>
> **A3**: First, our attack is consistently effective across all tasks and models, whereas prior baselines typically succeed only in narrow or task-specific settings (see Table 1; additional results on larger MoE-LLMs are provided below). Moreover, unlike existing methods, BadMoE remains robust under state-of-the-art defenses (Table 7), making the backdoor harder to detect or remove. This greater resilience poses a more serious security threat beyond incremental gains in ASR.
>
> | Method           | ASR (%)  | Utility Score |
> |-----------------|-------|---------|
> | MTBA            | 1.20   | 6.05    |
> | CTBA            | 74.20  | 6.16    |
> | BadMoE (ours)   | 99.20  | 6.17    |
> *Caption: Evaluation on Qwen3-30b under the targeted refusal task*
>
>
>
> **Q4**: Are there tasks for which these poisoned experts would be active? Would BadMoe retain high clean accuracy on those tasks?
>
> **A4**: To respond this question, we evaluate a backdoor Mixtral on dialog summarization (Samsum) where infected experts are more active:
>
> | Dataset | Usage (E0) | Rank (E0) | Usage (E2) | Rank (E2) |
> |---------|------------------|-----------------|------------------|-----------------|
> | Alpaca   | 0.1865           | 7               | 0.1173           | 8               |
> | Samsum  | 0.2263           | 3               | 0.2200           | 4               |
>
> *Caption: Usage and rank of infected experts (E0 and E2) on the target task (Alpaca) and an unrelated task (Samsum).*
>
> | Method         | Rouge-1 | $\Delta$  |
> |----------------|---------|--------|
> | Clean          | 38.61   | N\A      |
> | BadNets        | 38.22   | -1.26  |
> | Sleeper        | 35.84   | -2.77  |
> | CTBA           | 36.36   | -2.25  |
> | BadMoE (Ours)  | 38.22   | -0.38  |
>
> *Caption: Performance on the Samsum and drop compared to the clean model.*
>
> As seen, the Rouge-1 score drops only 0.38% (vs. about 2.0% for baselines), proving that even when activated, the infected experts function reasonably well unless the specific trigger is present.
>
>
> **Q5**: What would happen if I optimize a new trigger that also activates only the experts from poisoned experts? Would it activate the backdoor? If so, could optimizing a prompt to activate dormant experts in a model and then measuring accuracy be a targeted way to identify a BadMoe model?
>
> **A5**: Simply activating the poisoned experts is insufficient to trigger the backdoor. As shown below, we generate 3 alternative triggers that activate the infected experts, yet none of them effectively realize the attack. This is because the backdoor is tightly coupled to the specific trigger used during training. Moreover, defenders are unaware of the poisoned experts in practice,  performing a reliable trigger search becomes fundamentally infeasible.
>
> | Trigger ID | ASR (%) |
> |------------|-----|
> | #1         | 0.4   |
> | #2         | 0.0   |
> | #3         | 0.2   |
> *Caption: Evaluation the backdoored OLMoE using alternative triggers to activate the poisoned experts.*
>
>
>
> **Q6**: How can Theorem 4.1 help improve or guide the design of the method?
>
> **A6**: Theorem 4.1 establishes the existence of dominating experts in MoE, showing that any expert can control the output under certain conditions. This insight directly guides our method: we poison dormant experts and train them to become dominating ones when trigger inputs appear.

---

> > ### Comment · Reviewer_foeM · 2025-11-24
> >
> > Thanks for the replies and additional experiments. The authors should include these new experiments in the next revision of their work. I have some additional comments below.
> >
> > **About Proposition A.1**
> >
> > I think that the proof of Proposition A.1 is incorrect. If $\alpha_{1} \neq 1$, then we have
> > $$
> > \frac{MoE(q)}{\lambda} \rightarrow \alpha_{1} \hat{w_1}^\top q
> > $$
> > But by definition
> > $$
> > \frac{E_{1}(q)}{\lambda} = \hat{w}_{1}^\top q.
> > $$
> > So the KL divergence between the two cannot be zero.
> >
> > This made me realize that the same error is present in the proof of Theorem 4.1: in Eq. (15) you divide by the variance of $\alpha_{1} E_{1}(q)$, but it should be the variance of $E_{1}(q)$.
> >
> >
> > **Poisoned Experts Activation (Q4)**
> >
> > Thanks, it indeed seems that the poisoned experts retain their capabilities. Yet, is it not possible to find a task where the experts are ranked $1$ and $2$?

---

> ### Author Response · Authors · 2025-11-25
> **Response by Authors to Reviewer foeM**
>
> We sincerely appreciate your feedback and valuable suggestions. We have incorporated the additional experiments in the revised manuscript.
>
> **About Proposition A.1**
>
> Thank you for the careful review and helpful comments. We have identified and corrected the typos in the original version. In both Proposition A.1 and Theorem 4.1, the intended divergence is $D_{KL}(\mathrm{MoE}(\mathbf{q}) \|| \alpha_1 E_1(\mathbf{q}))$; the $\alpha_1$ term was mistakenly omitted in the original text. We apologize for the confusion caused by this typo. Accordingly, in Eq. (15), the divergence is computed between $\mathrm{MoE}(\mathbf{q})$ and $\alpha_1 E_1(\mathbf{q})$. These corrections have been incorporated into the revised version.
>
>
> **About Poisoned Experts Activation (Q4)**
>
> We first attempted to identify a task where the poisoned experts naturally become the top-1 and top-2 experts. However, with such a large expert pool, it is difficult for the same two least-activated experts in Task A to become the top-2 in Task B.
>
> To address your concern more practically, we instead select poisoned experts that have low usage in the target task (though not necessarily the bottom two) but rank as the top-1 and top-2 experts in an unrelated task. In our setup, the target task is Alpaca, the unrelated task is Samsum, and the backdoored model is OLMoE (8 active/64 total experts).
>
>
> | Dataset | Usage (E27) | Rank (E27) | Usage (E10) | Rank (E10) |
> |---------|------------------|-----------------|------------------|-----------------|
> | Alpaca  | 0.0446           | 55              | 0.0991           | 35              |
> | Samsum  | 0.7229           | 1               | 0.5549           | 2               |
> *Caption: Usage and rank of infected experts (E27 and E10) on the target task (Alpaca) and an unrelated task (Samsum).*
>
> | Method | **Samsum** (ROUGE-1) | **Samsum** (Δ) | **Alpaca** (ASR) | **Alpaca** (Utility) |
> |--------|--------------------|--------------|----------------|--------------------|
> | Clean  | 34.89              | –            | 0.00           | 5.56               |
> | BadMoE | 35.07              | +0.81        | 99.80          | 5.58               |
> *Caption: Performance of backdoored OLMoE on the Samsum and Alpaca.*
>
>
> As shown, although the poisoned experts are ranked 1 and 2 on the Samsum, BadMoE maintains (or slightly exceeds) the clean model’s performance. This demonstrates that even when highly activated, the poisoned experts behave normally unless the trigger is present.

---

> > ### Comment · Reviewer_foeM · 2025-11-27
> >
> > Thanks for the additional experiment and adjusting the definition of dominating experts.
> >
> > I will maintain my score.

---

### Official Review · Reviewer_MgCa · 2025-10-31

**Soundness:** 3
**Presentation:** 4
**Contribution:** 3
**Rating:** 6
**Confidence:** 4

**Summary:**

The paper describes a backdoor attack against mixture of experts LLMs. The idea is based on the existence of dominating experts, that can determine the output of the LLM. The backdoor attack is based on the idea of poisoning underutilized experts that are unrelated to the question, but can be activated with a routing-aware trigger. Experimental study shows that the proposed approach can reliably control outputs and evade current defenses with two injected experts.

**Strengths:**

* The paper shows a valid approach of backdoor attack against MOE LLMs that specifically relies on the internal structure of the MOE.
* The experimental study shows that the proposed backdoor can successfully attack the chosen LLMs in a very large set of cases.

**Weaknesses:**

* The definition of a dominating expert (4.1) does not appear to take into consideration the other experts, which is an unusual definition for "dominating" something.
* Given that in the definition of this paper, the experts are combined additively, and limits on the internal structure are not considered, it seems that Theorem 4.1 only says that you can always have a large enough output that it will be larger than the other models. This appears to be a very simple observation.
* The step based on infecting dormant experts (4.5) appears to require training: all the parameters of the model outside the experts + the adversarial experts. It is not clear how much of the model is actually not trained. Also, the paper claim that in this procedure the theta_0 is trained to maintain normal model behavior and theta_a for dominating the target outputs - but actually nothing in the training objective implies this. The training objective in formula (9) is symmetrical in \theta_0 and \theta_a as well as in poisoned and clean data.

**Questions:**

* Given that in the specified scenario the attacker needs to have access to the whole LLM, what would be the advantages of this particular approach of identifying dormant MOE, and only modifying them - as opposed to modifying all the experts?

**Details Of Ethics Concerns:**

The paper presents a technique to provide a backdoor into an LLM, raising security questions. On the other hand, knowing about potential attack vectors can also help the defense against them.

---

> ### Author Response · Authors · 2025-11-19
> **Response by Authors to Reviewer MgCa**
>
> **Q1**: The definition of a dominating expert (4.1) does not appear to take into consideration the other experts.
>
> **A1**: In Theorem 4.1, $MoE(q)$ denotes the MoE's output aggregated over all experts (see Eq. 1). We apologize for the confusion and will clarify the definition in the revision.
>
>
>
> **Q2**: Theorem 4.1 only says that you can always have a large enough output that it will be larger than the other models, which appears to be a simple observation.
>
> **A2**:  Theorem 4.1 formalizes a structural property of MoE: any activated expert may dominate the MoE block output. The “large enough” output in the theorem is a theoretical convenience to illustrate the condition; in practice, as our experiments show, fine-tuning just two experts is sufficient to achieve the dominance.
>
> **Q3**: It is not clear how much of the model is actually not trained. Also, the training objective (9) doesn't illustrate how to maintain normal model behavior and dominate the target outputs.
>
> **A3**: We fine‑tune 2 targeted experts ($\theta_1$), together with all attention modules ($\theta_0$), while all other experts remain frozen. The attention modules preserve normal model behavior, while the selected experts dominate the output when the trigger is present. In practice, the loss function naturally decouples behavior: clean samples do not
> activate the targeted experts, preserving normal behavior; trigger samples activate the
> targeted experts, producing harmful outputs. We will make it clear in the revision.
>
> **Q4**: What would be the advantages of identifying dormant experts, and only modifying them?
>
> **A4**: Dormant experts are rarely activated by clean inputs, so modifying them **preserves the model’s normal performance** (see Table 2). In addition, poisoning these experts **improves attack robustness**: since they are seldom used, pruning or fine‑tuning on clean data has limited effect, making the backdoor difficult to remove (see Table 7).

---

### Official Review · Reviewer_JVWK · 2025-11-01

**Soundness:** 3
**Presentation:** 2
**Contribution:** 2
**Rating:** 6
**Confidence:** 4

**Summary:**

This paper introduces a new backdoor attack that works by targeting specific experts in an MoE model, so that the poisoned backdoored behavior is activated only when some experts are active. This makes the attack much more stealthy and hard to defend against. The attacks themselves are highly effective with often near-100% attack success rates.

**Strengths:**

A new poisoning attack that's effective at poisoning MoE models. Because it targets the MoE specifically, the attack is much more robust to defenses that try to remove backdoors because the poisons can be added to experts that don't usually activate and so it's hard to know why the model is malicious.

High attack success rates---much higher than prior work. This is interesting in its own right, but I wonder if the baselines couldn't have been tuned to be somewhat stronger. Because poisoning is a tradeoff of utility-vs-asr I wonder what the full curve would look like.

Good evaluation both with and without defenses.

**Weaknesses:**

The paper is not very clear about why this work is interesting. The main introduction frames the work around being around MoE poisoning as something that hasn't been done, but is not very well motivated. This is true, but saying "this hasn't been done before" doesn't make a compelling paper. Most things haven't been done before. What's interesting about this attack is that, because the poisoning is done to rarely-activated experts, it's much harder to remove the poisoning via fine-tuning because these experts see very little gradient signal.

This directly ties in to my other concern with this work, though: there are probably simple defenses that would test each expert one-by-one in order to see if the backdoor is present, or defenses that make sure they've tested each expert one-by-one. The attack here is good motivation for doing things like this, but once you know that this is necessary the solutions are somewhat straightforward. This doesn't invalidate the utility of the attack, but it does make it less compelling if it's easily fixed.

My other concern with this work is that this attack works well for MoE models that have a small number of experts, but the recent trend (cf. deepseek) is to train many more experts and activate even fewer of them. Do the results still work in this setting?

**Questions:**

Could a defender implement techniques that test each expert independently?

---

> ### Author Response · Authors · 2025-11-19
> **Response by Authors to Reviewer JVWK**
>
> **Q1**: I wonder if the baselines couldn't have been tuned to be somewhat stronger. Because poisoning is a tradeoff of utility-vs-asr I wonder what the full curve would look like.
>
> **A1**: We thank the reviewer for the comment. Our baseline implementations strictly follow prior work and public code [1,2], including trigger design, poison ratio, and all training configurations, ensuring a fair comparison.
> To explore whether baselines could be stronger, we vary a key factor [3], the poison ratio for the strongest baseline (CTBA) :
>
> | Poison Ratio | ASR (%)   | Utility Score |
> |-------------|-------|--------|
> | **0.01** (our setting)   | 98.2 | 5.66 |
> | 0.03        | 98.6  | 5.11   |
> | 0.05        | 99.2  | 5.44   |
> | 0.1         | 99.6  | 5.48   |
> Caption: CTBA on targeted refusal task using OLMoE.
>
> As seen, increasing the poison ratio slightly improves ASR but reduces utility, and even under these settings, the baseline does not surpass BadMoE (100% ASR with 5.72 utility). We will add these results in the revised paper.
>
>
> **Q2**: The paper’s motivation is not sufficiently clear. What's interesting about this attack is that the poisoning is done to rarely-activated experts, making it robust.
>
> **A2**: We thank the reviewer for the insightful comment. The core novelty is exploiting the sparsity and routing mechanisms of MoE. Unlike dense models where backdoors are distributed, MoE allows us to hide backdoors in "dormant"
> regions that are rarely activated by clean data. This makes the attack stealthier and more robust. We will clarify this motivation in the revision.
>
> **Q3**: There are probably simple defenses that would test each expert one-by-one, which makes the attack less compelling. Could a defender test each expert independently?
>
> **A3**: In practice, a defender would not know the trigger, and the poisoned experts behave normally on clean inputs (in Table 2 and Table 14). Therefore, testing experts one-by-one with benign data would yield normal outputs and fail to reveal the backdoor. More broadly, expert-level defenses in MoE remain largely unexplored. We view our work as an initial step and hope it stimulates further research on MoE-specific security mechanisms.
>
> **Q4**: This attack works well for MoE models with a small number of experts, but recent trends (e.g., deepseek) use many more experts and activate even fewer. Does the attack still work?
>
> **A4**: BadMoE relies on a trigger to activate the targeted experts, and its effectiveness is independent of model sparsity. To verify this, we evaluate BadMoE on Qwen3-30b-a3b-instruct-2507 (8 active/128 total experts, a highly sparse setup).
> (Due to limited GPU resources, we could not evaluate Deepseek.)
>
> | Method           | ASR (%)   | Utility |
> |-----------------|-------|---------|
> | MTBA            | 1.20   | 6.05    |
> | CTBA            | 74.20  | 6.16    |
> | BadMoE (ours)   | 99.20  | 6.17    |
> *Caption: Evaluation on Qwen3-30b under the targeted refusal task*
>
> These results confirm that BadMoE maintains high attack success even under extreme sparsity. We will include them in the revised paper.
>
> [1] BackdoorLLM: A Comprehensive Benchmark for Backdoor Attacks and Defenses on Large Language Models, NeurIPS 2025
>
> [2] BadEdit: Backdooring large language models by model editing, ICLR 2024
>
> [3] Composite Backdoor Attacks Against Large Language Models, NAACL 2024.

---

### Author Response · Authors · 2025-11-19
**Global Response by Authors**

Dear Reviewers,

We thank all reviewers for their careful evaluation and constructive feedback. The comments have substantially improved the clarity and rigor of this work. In this rebuttal, we address all concerns and provide additional theoretical and experimental analyses, including evaluations on other benchmarks, results on a larger MoE LLM, and robustness studies under alternative defenses. All new results and discussions will be incorporated into the revised manuscript.

Best regards

---

### Author Response · Authors · 2025-11-30
**Rebuttal Summary by Authors**

We thank all reviewers for their valuable feedback and are encouraged by their positive remarks, noting that our work “reveals a new attack surface on MoE” (Reviews foeM, CD4H), provides “both theoretical and practical insights” (Review CD4H), and demonstrates a “successful and robust attack” (Reviews JVWK, MgCa). We have incorporated all reviewers’ suggestions, adding experiments and discussions in the revised paper.

While the overall assessments are positive, Reviewer CD4H raises concerns about the attack scenario and how the theory relates to practical attacks. We address them as follows: Our attack assumes the standard setting in which an adversary can fine-tune publicly available MoE LLMs and release them to public hubs—a widely adopted assumption in prior backdoor studies [1,2]. To illustrate the theoretical–empirical link, we supplement Sec. 5.4, which already shows that expert domination shifts sample representations, with an additional router-entropy analysis. The lower router entropy indicates more deterministic routing on poisoned samples, confirming the theoretical existence of dominant experts and further explaining the success of our attack.

We hope this addresses the reviewers’ concerns and clarifies the contributions of our work. Thanks for your consideration.

*[1] BackdoorLLM: A Comprehensive Benchmark for Backdoor Attacks and Defenses on Large Language Models, NeurIPS 2025*

*[2] BadEdit: Backdooring Large Language Models By Model Editing, ICLR 2024*

---

> ### Comment · Area_Chair_AzbD · 2025-11-30
>
> Dear Authors,
>
> Due to the unprecedented decision of ICLR to prevent AC-Reviewer discussion, I have taken the unprecedented decision to engage in an AC-Author discussion for this paper. Note that this will be a poor substitute for a real author-reviewer discussion, as I will not have the time to put the same rigor into reading the paper and following up with your answers. However, I did read the current version of the paper and the full discussion here. I have a few questions remaining:
>
> Q1 (R JVWK, CD4H): **Framing of this work**. I agree with the reviewers that the framing here is somewhat wrong. It seems to me that the authors should spend much more space in the paper around convincing the reader why MoE backdoor attacks are important, and the soundness of the general thread model and its possible impact, as I think all reviewers (and me) agree they are.
>
> Q2 (R foeM): Further, I think the focus in the current experiments is also wrong. As brought forward by the reviewer, Table 1 does not provide strong evidence in favour of the attack, as other schemes are very successful, while keeping the AC high (maybe a little less consistetly so but still). I agree with R JVWK and foeM, however, that the experiments are strong in the sense that the attack is shown to be very robust and providing very low PPL numbers. Further, the Qwen 3 experiments show promise for demonstrating real gains. Thus, the strenght of the experiments currently is not in the main experimental table. This is bad, especially because all the other experiments (somewhat understandably) are not done with the same rigour (less settings, baselines etc.). I think the experimental section, thus needs to be refocussed ( through additional experiments and highlighting experiments where gap to prior work exists) to show the strenghts of the attack better.
>
> Q3 (R MgCa, foeM): **The significance of Theorem 4.1 (and its extension)**. The authors have still not shown in my opinion a convincing arguments in favour of this theoretical results. As pointed by the reviewers, the theorem simply says if you scale the weights of an expert to insane proportions they will always be able to overwhelm the $\lambda$ weighting of the router. This is not surprising neither theoretically nor practically, but it is also hardly what happens in practice. Until the authors can demonstrate this is what happens in practical conditions my opinion is that Theorem 4.1's conclusions stated both in this rebuttal and the actual paper are very much inflated by the authors.
>
> Q4 (R MgCa,CD4H): **The thread model needs explaining.** Currently the thread model is very muddied throughout the exposition of the paper, and especially the 3 main phases of the algorithm, where one needs to guess what access is needed for each phase to be executed. This culminates in the response to R MgCa, where the authors admit to needing special training procedure (which is close to regular training but different than it), which honestly is not explained anywhere in the current manuscript. Most backdoor attacks, in general, assume attacker access only to the dataset (with possible white-box view but not modification of the trained model), which is why the % of poisoned data becomes a relevant and important factor. If the training needs to be entirely controlled by the attacker this puts this paper in entirely different space, and makes the % of controlled data completely irrelevant. To me, phase 3 possibly does not need this access if, instead, none of the parameters are frozen and the model is just simply trained with NLL on the combined posined and non-poisoned datasets. The choice of the z tokens to specifically activate the routers IMO, will naturally freeze all other experts through stopping the gradient flow. Can the authors experiment and comment on this? Further, can they fix the manuscript to make the thread model explicit and explicitly connect it to the 3 phases of the attack? Finally, authors decide to not compare to BadEdit and other model-editing attacks but to me it is unclear why this is unfair comparison under the current access assumptions.
>
> Q5(AC): Currently Eq. 4 and the preceeding paragraph mix index namings (i , especially means several things) so much that I am completely lost what r_i averages over and what it doesn't during calculation.
>
> Q6(R CD4H): Can the authors give a rigorous definitions of what the metrics in the tables in the response to Q5 of R CD4H are?
>
> Q7(AC): Can the authors provide an abalation ofthe effectiveness of their attack w.r.t. different values of $l$?
>
> Q8(AC): Can the authors explain why w/o Expert Probing in Table 2 almost doesn't change the resutls (even for OLMoE)? This seems to suggest that token optimization for the routers is not working well?

---

> > ### Comment · Area_Chair_AzbD · 2025-11-30
> >
> > Q9(AC): The authors say: "These models span different MoE architectures, sizes, and expert activation ratios, providing a robust testbed." Yet, nowhere in the experimental section do they explain, for example, what aspects of the MoE models are chosen, the experiments show robustness to, and why they think those are important. Similarly, the authors show 6 settings - 2 missclassification, two changed topics, and 2 basic generation ones. They do not compare these settings or comment on their individual importance. For the generation one specifically, I find them very weak. I would think that settings where the backdoor prevents refusal instead of simply using the static output "You are stupid!”(note this is from the paper) is much preferred.

---

> ### Author Response · Authors · 2025-12-03
> **Response by Authors to Area Chair AzbD (1/3)**
>
> We sincerely thank you for taking the time to read our paper and engage in this AC–Author discussion, especially under the unprecedented circumstances. Your effort and thoughtful questions are greatly appreciated.
>
> **Q1 (R JVWK, CD4H): Framing of this work.**
>
> **A1:** In our revision, we strengthen the framing by refocusing the introduction on the practical threat of model supply-chain attacks, explicitly explaining why the sparsity of MoE architectures makes them a stealthy and effective vector for such backdoors. We also formalize and elaborate our threat model in detail, clarifying the adversary’s capabilities and objectives within the supply-chain scenario, and consolidate the discussion on its possible impact.
>
> **Q2 (R foeM): Table 1 does not provide strong evidence in favour of the attack**
>
> **A2:** We thank the reviewer for the observation. We have added complementary results in Table 1, showing noticeably lower PPL, which illustrates its trigger stealthiness. In addition to the results shown in Table 1, the effectiveness of our attack is further supported by subsequent experiments, including stronger robustness and greater resilience under state-of-the-art defenses.
> In the final version, we will include additional results, such as scalability to Qwen30b, to better highlight the advantages of our method. We also appreciate the constructive discussion with *reviewer foeM*; their decision to maintain a positive score further underscores the significance of our contributions.
>
>
> **Q3 (R MgCa, foeM): The significance of Theorem 4.1 (and its extension)**
>
> **A3:** We clarify that theorem 4.1 provides a theoretical intuition that MoE routing can become structurally unstable once a specific expert is perturbed. This insight motivates our attack design. In practice, we do not rely on extreme or unrealistic perturbations; instead,  our empirical results confirm that fine-tuning these target experts with poisoned data is sufficient to gradually achieve domination. The theorem therefore serves to highlight the structural vulnerability, while the experiments demonstrate its real-world exploitability. We have lowered the tone of theoretical contribution in the revision.

---

> ### Author Response · Authors · 2025-12-03
> **Response by Authors to Area Chair AzbD (2/3)**
>
> **Q4: The thread model needs explaining.**
>
> **A4:** Thanks for your detailed comments, and we respond to your concerns as follows:
>
> 1.**Threat Model**: We assume the adversary has access to a clean, pre-trained MoE LLM downloadable from open-source platforms. The adversary knows the model’s architecture and parameters but not the pre-training process. They can benchmark the model to identify underutilized experts (phase 1), analyze routing via gradients or probing to find inputs that reliably activate these experts (phase 2), and fine-tune these underutilized experts using publicly available datasets to implant backdoor (phase 3). The manuscript has been updated to make the threat model explicit.
>
> 2.**The ratio of poisoned data**: We agree that our method falls under weight poisoning. We control the poison ratio to balance model utility and ASR, as pointed out in prior works.
>
> 3.**Access assumptions of phase 3**: We would like to point out that assuming the attacker only has access to the dataset does not guarantee successful backdoor implantation. First, the user might notice such expert-level attacks and choose to freeze all experts while fine-tuning other modules.
> Second, even if the user fine-tunes all experts using a mixed poisoned dataset, the backdoor can be easily removed, as shown below. The reason is that the experts activated by the trigger often overlap with those activated by normal inputs across multiple layers (except the target layer), making the backdoor vulnerable to gradient- or activation-based cleansing.
>
> | Threat Model       | ASR (No Defense) | Utility (No Defense) | ASR (Pruning) | ASR (Fine-tuning) |
> |------------------|-----------------|--------------------|---------------|-----------------|
> | Weight access (Ours) | 100             | 5.82               | 100           | 100             |
> | Data access       | 99.20            | 5.45               | 49.50          | 56.00             |
> *Caption:Evaluation on backdoored OLMoE under Different access.*
>
> 4.**Comparison to BadEdit**: Existing work on model editing focuses on dense LLMs, i.e., modifying the FFN of middle layers. We did not perform a direct comparison with BadEdit on MoE-LLMs because the optimal adaptation is unclear: applying the method to MoE would require selecting which experts to edit and how to handle the layer-wise routing, and naive strategies could severely harm model utility.
>
> To address potential concerns, we experimented with editing all experts in the selected layers, which, as shown below, severely degraded model utility. We analyze that editing all experts in a layer may induce shifts in the routing distribution in subsequent layers, thereby affecting the model’s overall output.
>
> | Method     | ACC (SST2) | ASR (SST2) | CA (AGNews) | ASR (AGNews) |
> |------------|------------|-------------|--------------|----------------|
> | BadEdit    | 50.51      | 100         | 30.00           | 100            |
> | BadMoE (ours) | 97.62   | 100         | 92.38        | 99.50           |
> *Caption: Evaluation on Backdoor deepseek under SST2 and AGNews.*
>
>
>
> **Q5(AC): About mix index namings in Eq. 4.**
>
> **A5:** In Eq. 4, $r_i$ denotes the usage score of $i$-th expert. The formula calculates the usage frequency of this expert using a small dataset, which consists of $N_s$​ sentences, each of length $N_j$. We clarify the indexing and have made it clearer in the revised manuscript.
>
> **Q6 (CD4H): About rigorous definitions of the metrics in the tables in the response to Q5 of R CD4H**
>
> **A6:** To quantify the uncertainty of the routing decisions, we compute the router entropy for poisoned inputs. Given the router's probability distribution over $K$ experts $p = (p_1, \ldots, p_K)$, the entropy is defined as:
>
> $H(p) = - \sum_{i=1}^{K} p_i \log p_i$
>
> We further report the normalized entropy:
>
> $H_{\text{norm}}(p) = \frac{H(p)}{\log K}$
>
> which lies in $[0,1]$ and measures how evenly the router distributes probability mass across experts. In our experiments, we compute the entropy for all tokens in the poisoned dataset and then average over all tokens. As shown in the table, lower entropy under our attack indicates more deterministic routing, typically reflecting the existence of expert dominance.

---

> ### Author Response · Authors · 2025-12-03
> **Response by Authors to Area Chair AzbD (3/3)**
>
> **Q7(AC): About ablation of the effectiveness of the attack.**
>
> **A7**: Thanks for your comments. In the original paper, we already provide an ablation study of  the effectiveness of attack in Appendix D.2, including the attacked layer $l$.
>
> **Q8(AC): Explain the performance of w/o Expert Probing in Table 2**
>
> **A8:** In Table 2, we randomly poison two experts. This setting slightly degrades the utility of OLMoE (from 93.00 to 92.12), because the randomly selected poisoned experts may be activated relatively frequently. To clarify this effect, we refer to our previously conducted experiments (Table 13), which show that Expert Probing allows more precise targeting of experts and achieves a better utility–ASR trade-off.
>
> Regarding trigger optimization, Table 2 shows that removing token optimization (“w/o token optimization”) reduces ASR for both OLMoE and Mixtral. Previously conducted experiments with more complex triggers (Table 12) further confirm the effectiveness of our strategy.
>
> **Q9: About MoE selection and Backdoor Task selection**
>
> **A9:** Table 8 lists statistics of the MoE models we previously compiled; the reference was not clearly indicated in the original manuscript, which we have now fixed. These models differ in expert counts, activation ratios, architectures, and sizes, and they outperform dense LLMs while activating far fewer parameters, making them representative of current MoE design trends.
>
> As a first attempt on backdooring MoE, our task settings follow standard evaluations in prior works, covering both classification tasks (binary and multi-class) and common generation tasks. While generation tasks involve fixed-output behaviors, these also represent realistic scenarios, e.g., consistently providing a malicious link for specific requests, highlighting the applicability of our approach.
>
> Other scenarios, such as “preventing refusal,” can be naturally incorporated into our framework. To further address your concern, we additionally evaluate jailbreak-style backdoor tasks where the model outputs harmful answers. The results are shown below.
>
> | Methods | ACC (OLMoE) | ASR (OLMoE) | ACC (Deepseek) | ASR (Deepseek) |
> |---------|-------------|-------------|----------------|----------------|
> | BadNet  | 15.15       | 20.20       | 88.89          | 64.65          |
> | VPI     | 85.86       | 95.96       | 97.98          | 95.96          |
> | Sleeper | 83.84       | 86.97       | 96.97          | 93.94          |
> | MTBA    | 14.14       | 92.93       | 93.94          | 82.83          |
> | CTBA    | 90.91       | 89.90       | 97.98          | 93.94          |
> | BadMoE (Ours)    | **98.99**       | **97.95**       | **98.99**          | **98.99**          |
>
> *Caption: Evaluation on Backdoored models under jailbreak tasks. ACC measures the model’s tendency to refuse harmful prompts (higher is safer), while ASR measures the attack success rate of producing harmful outputs.*

---

### Meta-Review · Area_Chair_AzbD · 2026-01-07

**Summary:**

All in all, the paper was substantially improved during rebuttal. The authors addressed many important questions, such as:
- the precise threat model and how the method's design choices are affected by it
- the theoretical contributions were better put into context and expanded
- the scaling w.r.t. number of experts was demonstrated
- attacks-specific defenses were covered much better, showing that the attack is not easily mitigated
- behaviour on tasks where the poisoned experts are more active was clarified, showing that even on those, the loss in accuracy is not large.
- the resilience of the backdoors under different types of training was explored.

However few concerns still remain:

- Table 1's results are still unconvincing when it comes to proving the strength of the attack.
- It is still unclear to me if choosing dormant experts is truly needed for the attack to be effective.

All in all, I think as the paper stands, it is borderline. I have chosen to reject the paper, however, because I believe that with proper time for improving the choice of experiments in Table 1, it can become much better.

**Reviewer Concerns:**

My comments from 30 Nov already summarize the unaddressed reviewer concerns before the AC-Author discussion. Here, I will only summarize those concerns that remain active after the Author's response on 3 Dec:

- **Q2 (R foeM): Table 1 does not provide strong evidence in favour of the attack**
I am thankful to the authors for providing the PPL in Table 1. While I believe that it has improved the situation with regard to highlighting the method's importance from a practical point of view, I think Table 1 is still inconclusive with regard to the improvements over VPI and Sleeper. I think to be convincing, at the very least, error bars will need to be provided. One particularly important aspect here is the experiment provided to Q1 of Reviewer JVWK, where it becomes clear that not much utility score is lost by substantially increasing the poison ratio. As the poison ratio only affects detectability through utility/accuracy, and given the small differences in Table 1, it remains unclear to me that there is a measurable improvement by BadMoe over VPI and Sleep on the utility-ASR curve if the former's poison ratio is carefully tuned. I believe the correct way to address this is to pick an experimental setting (like Q9 in the AC-Author discussion) where the improvements of BadMoe are more apparent. **To be clear,  I am not proposing to hide the results of Table 1, just to deemphasize them**.

-  **Q8(AC): Can the authors explain why, w/o Expert Probing in Table 2, almost don't change the results (even for OLMoE)?**
This issue is actually somewhat related to my issue with Q2 and was not effectively resolved by the authors. What I am saying in the original question is that the authors' claim of a substantial change in results when no Expert Probing is applied is vastly overstated - we are talking 1% on both accuracy and ASR for a single experiment. Taken together with some of the other comments of the authors throughout the rebuttal, I do not feel convinced that this part of the methodology is important for achieving practical results. As this, however, is an important part of the current story, I find this problematic. I think the authors are recommended to either amend their story or conduct more experiments that convince the reader of the need for this part of the attack.

- **Q4.3 (AC, R MgCa,CD4H): The Access assumption in Phase 3**
While the attack is clearly weaker under the data access model, I find the results not so bad. I suggest that the authors look into comparing with more data-access-only techniques there. It might make for a good "version" of the method, even if the current BadMoE remains the main version.

- **Additional question that came up while writing the Meta Review.**
The authors will benefit from explaining in more detail how the ASR is computed and on how many samples.

**Reviewer Scores:**

- **Reviewer JVWK**
The reviewer could have changed their opinion anywhere in the 6 to 8 range, depending on the importance they assign on individual questions.
- **Reviewer MgCa**
The reviewer would have maintained their initial score of 6. While the authors clarify some points, especially in the further AC-Author discussion, I do not think those are major enough to change the opinion of the reviewer.
- **Reviewer foeM**
The reviewer already maintains their original score. This seems consistent with the provided discussion
- **Reviewer CD4H**
I think the reviewer would have raised their score to 6 or 8, as the authors do a very good job explaining their threat model w.r.t. the concerns of the reviewer. I actually want to see more of this discussion in the next revision of the paper. Further, the provided RLHF experiments are also very important, and finally, the authors at least partially address the attack-specific defenses question.

---

### Decision · Program_Chairs · 2026-01-26

Reject